# The Uso1 globular head interacts with SNAREs to maintain viability even in the absence of the coiled-coil domain

Ignacio Bravo-Plaza[1], Victor G Tagua[2], Herbert N Arst[3], Ana Alonso[1], Mario Pinar[1], Begoña Monterroso[4], Antonio Galindo[5], Miguel A Peñalva[1]*

[1]Department of Cellular and Molecular Biology, CSIC Centro de Investigaciones Biológicas, Madrid, Spain; [2]Instituto de Tecnologías Biomédicas, Hospital Universitario Nuestra Señora de Candelaria, Santa Cruz de Tenerife, Spain; [3]Department of Infectious Diseases, Faculty of Medicine, Flowers Building, Imperial College, London, United Kingdom; [4]Department of Structural and Chemical Biology, CSIC Centro de Investigaciones Biológicas, Madrid, Spain; [5]Division of Cell Biology, MRC Laboratory of Molecular Biology, Francis Crick Avenue, Cambridge, United Kingdom

*For correspondence:
penalva@cib.csic.es

Competing interest: The authors declare that no competing interests exist.

**Abstract** Uso1/p115 and RAB1 tether ER-derived vesicles to the Golgi. Uso1/p115 contains a globular-head-domain (GHD), a coiled-coil (CC) mediating dimerization/tethering, and a C-terminal region (CTR) interacting with golgins. Uso1/p115 is recruited to vesicles by RAB1. Genetic studies placed Uso1 paradoxically acting upstream of, or in conjunction with RAB1 (Sapperstein et al., 1996). We selected two missense mutations in *uso1* resulting in E6K and G540S in the GHD that rescued lethality of *rab1*-deficient *Aspergillus nidulans*. The mutations are phenotypically additive, their combination suppressing the complete absence of RAB1, which emphasizes the key physiological role of the GHD. In living hyphae Uso1 recurs on puncta (60 s half-life) colocalizing partially with the Golgi markers RAB1, Sed5, and GeaA/Gea1/Gea2, and totally with the retrograde cargo receptor Rer1, consistent with Uso1 dwelling in a very early Golgi compartment from which ER residents reaching the Golgi recycle back to the ER. Localization of Uso1, but not of Uso1[E6K/G540S], to puncta is abolished by compromising RAB1 function, indicating that E6K/G540S creates interactions bypassing RAB1. That Uso1 delocalization correlates with a decrease in the number of Gea1 cisternae supports that Uso1-and-Rer1-containing puncta are where the protein exerts its physiological role. In S-tag-coprecipitation experiments, Uso1 is an associate of the Sed5/Bos1/Bet1/Sec22 SNARE complex zippering vesicles with the Golgi, with Uso1[E6K/G540S] showing a stronger association. Using purified proteins, we show that Bos1 and Bet1 bind the Uso1 GHD directly. However, Bet1 is a strong E6K/G540S-independent binder, whereas Bos1 is weaker but becomes as strong as Bet1 when the GHD carries E6K/G540S. G540S alone markedly increases GHD binding to Bos1, whereas E6K causes a weaker effect, correlating with their phenotypic contributions. AlphaFold2 predicts that G540S increases the binding of the GHD to the Bos1 Habc domain. In contrast, E6K lies in an N-terminal, potentially alpha-helical, region that sensitive genetic tests indicate as required for full Uso1 function. Remarkably, this region is at the end of the GHD basket opposite to the end predicted to interact with Bos1. We show that, unlike dimeric full-length and CTRΔ Uso1 proteins, the GHD lacking the CC/CTR dimerization domain, whether originating from bacteria or *Aspergillus* extracts and irrespective of whether it carries or not E6K/G540S, would appear to be monomeric. With the finding that overexpression of E6K/G540S and wild-type GHD complement *uso1Δ*, our data indicate that the GHD monomer is capable of providing, at least partially, the essential Uso1 functions, and that long-range tethering activity is dispensable. Rather, these findings strongly suggest that the essential role of Uso1 involves the regulation of SNAREs.

## Editor's evaluation

This valuable manuscript explores the role of Uso1/p115, a protein that has been implicated in vesicle tethering at the ER-Golgi interface. By investigating a Uso1 mutant that allows Aspergillus cells to survive in the absence of Rab1, the authors conclude that the essential role of Uso1 is not actually tethering, but rather SNARE complex assembly mediated by the globular head domain. This convincing analysis significantly advances our understanding of Uso1 and also prompts a reevaluation of long-standing assumptions about coiled-coil proteins involved in vesicular transport.

## Introduction

Vesicular traffic at the ER/Golgi interface is the cornerstone of the secretory pathway (*Barlowe and Miller, 2013*; *Weigel et al., 2021*). In current models, in which traffic across the Golgi is driven by cisternal maturation (*Day et al., 2013*; *Pantazopoulou and Glick, 2019*), COPII vesicles generated at specialized domains of the ER fuse homotypically and heterotypically to form and feed the earliest Golgi cisternae (*Rexach et al., 1994*). As straightforward as this step might seem, it involves a sophisticated circuitry of regulation. Actual fusion is in part mediated by compartmental-specific sets of four-membered SNARE protein complexes (SNARE bundles) (*Malsam and Söllner, 2011*; *Pelham, 2001*; *Rizo and Südhof, 2012*). Most SNARES are type II single TMD proteins, whose N-terminal cytosolic domain contains nearly all the polypeptide, except a few lumenal residues. Like any other transmembrane proteins, SNAREs are synthesized in the ER. This implies that they have to travel to compartments of the cell as distant as the plasma membrane in a conformation that precludes them from catalyzing what would be a calamitous fusion of non-cognate donor and acceptor compartments. Achieving the strictest specificity is particularly challenging in the ER-to-Golgi stage that, as the first step in the secretory pathway, represents an obligate point of transit for each and every transmembrane SNARE. Therefore, the only SNARES acting in this first step are the Qa Sed5, the Qb Bos1, the Qc Bet1, and the R-SNARE Sec22, which form the bundle mediating fusion of carriers that coalesce into cisternae (*McNew et al., 2000*).

Given the central role played by the secretory pathway in the physiology of every eukaryotic cell, it is unsurprising that this step involves regulatory factors which are essential for cell survival. One is the SM (Sec1, Munc-18) protein Sly1, which promotes SNARE bundle formation (*Bracher and Weissenhorn, 2002*; *Peng and Gallwitz, 2002*; *Thomas et al., 2019*) Another is the TRAPPIII complex, which interacts with the external coat of COPII carriers and acts as a guanine nucleotide exchange factor (GEF) for RAB1 (*Bracher and Weissenhorn, 2002*; *Cai et al., 2007*; *Galindo et al., 2021*; *Joiner et al., 2021*; *Lord et al., 2011*; *Peng and Gallwitz, 2002*; *Pinar and Peñalva, 2020*; *Riedel et al., 2018*; *Thomas et al., 2018*; *Thomas et al., 2019*). This small GTPase is a key player that transiently recruits protein effectors from the cytosol to donor and acceptor membranes (*Søgaard et al., 1994*) and regulates SNARE assembly through an as yet undefined mechanism (*Lupashin and Waters, 1997*; *Sapperstein et al., 1996*). One RAB1 effector is a fungal protein denoted Uso1, whose highly conserved metazoan homolog is p115. These are homodimers with a globular N-terminal head and a long C-terminal coiled-coil region characteristic of tethering proteins, which bring donor and acceptor membranes into the distance at which v- and t-SNAREs can engage into the productive *trans*-SNARE complex that mediates membrane fusion (*Cao et al., 1998*; *Nakajima et al., 1991*; *Sapperstein et al., 1996*; *Sapperstein et al., 1995*; *Seog et al., 1994*; *Yamakawa et al., 1996*).

Despite most Golgi tethers are non-essential, Uso1 is unique in that it is required for viability. *uso1-1*, an *S. cerevisiae* amber mutation truncating most, but not all the coiled-coil region is viable, yet further upstream truncation removing the complete coiled-coil is lethal, which was taken as evidence that tethering is the essential function of Uso1 (*Seog et al., 1994*). In addition, as the coiled-coil is predicted to mediate dimerization, it is broadly accepted that Uso1 is 'just' an essential homodimer that tethers vesicles to the acceptor membrane. However, genetic evidence stubbornly indicates that Uso1 plays additional functions related to SNAREs. For example, *Sapperstein et al., 1996* showed that SNAREs function downstream of Uso1. Notably, the view that Uso1 is a mere RAB1 effector was challenged by the observation that overexpressing Ypt1 (yeast RAB1) rescues the lethality of *uso1Δ*, whereas the reciprocal is not true, indicating that Uso1 acts upstream of or in conjunction with RAB1 (*Sapperstein et al., 1996*).

Our laboratory is interested in deciphering the domains of action of RAB GTPases in the genetic and cell biological model organism *Aspergillus nidulans* (*Pinar and Peñalva, 2021*). We have previously gained mechanistic insight into the activation of RAB11 by TRAPPII by exploiting a forward genetic screen for mutations bypassing, at the restrictive temperature, the essential role of the key TRAPPII subunit Trs120 (*Pinar et al., 2019*; *Pinar et al., 2015*; *Pinar and Peñalva, 2020*). In this type of screen, a strain carrying a temperature-sensitive (*ts*) mutation in the gene-of-interest is mutagenized and strains bypassing lethality at the restrictive temperature are identified and characterized molecularly. A well-characterized, conditionally lethal *rab1* mutation is available (*Pinar et al., 2013*), enabling us to investigate pathways collaborating with RAB1 in anterograde traffic. We isolated two *uso1* missense mutations causing substitutions in the GHD. When combined together, these rescued the lethality resulting from *rab1Δ* and promoted the localization of the protein to early Golgi cisternae by increasing Uso1 binding to the cytosolic region of the Qa SNARE Bos1. Importantly, we show that endogenous expression of a protein consisting solely of the double mutant GHD, or overexpression of double mutant or wild-type GHD, rescues the lethality resulting from *uso1Δ*, even though data strongly suggest that the GHD is monomeric. Our results show that one essential role of RAB1 is recruiting Uso1 to membranes, and strongly suggest that the essential role of Uso1 is not tethering membranes, but rather regulating the formation of the cognate SNARE bundle, almost certainly placing Uso1 as a component of the SNARE fusion machinery.

## Results

### Missense mutations in *uso1* rescue the lethality resulting from *rab1Δ*

*rab1^A136D* (hereby *rab1^ts*) mutants do not grow at 37 °C. However, when we plated UV-mutagenized conidiospores of the *rab1^ts* mutant at this temperature, we obtained colonies showing different degrees of growth, presumably carrying mutations rescuing the lethality resulting from *rab1^ts*. One was chosen for further characterization. By sexual crosses and parasexual genetics, this strain was shown to carry a single suppressor mutation, denoted *su1rab1^ts*, that co-segregated with chromosome VIII. Meiotic mapping narrowed *su1rab1^ts* to the vicinity (2 cM) of *hisC*. 40 kb centromere distal from *hisC* lies AN0706 (*Figure 1A*) encoding *Aspergillus nidulans* Uso1, a conserved effector of RAB1. Sanger sequencing revealed the presence of a G16A transition (denoted E6K) resulting in Glu6Lys substitution in the *uso1* gene of *su1rab1^ts*.

To determine if the remaining suppressor strains were allelic to *su1rab1^ts*, we sequenced *uso1* from a further 13 isolates (*Figure 1B*). Of these, four were *rab1^ts* pseudo-revertants that had acquired a functionally acceptable mutation in the altered codon, and eight carried *uso1* E6K, suggesting that the screen was close to saturation. However, one mutation was found to be a different missense allele, *su85rab1ts* (denoted G540S) resulting in Gly540Ser substitution. Single mutant strains carrying these *uso1* mutations showed no growth defect, indicating that E6K and G540S were unlikely to result in loss-of-function, and suggesting instead that mutant strains had acquired features that made them largely independent of RAB1. These findings were unexpected because in *Saccharomyces cerevisiae* overexpression of Uso1 does not rescue the lethality of *ypt1Δ* mutants (Ypt1 is the yeast RAB1 homolog) (*Sapperstein et al., 1996*).

To demonstrate that *uso1^E6K* and *uso1^G540S* were causative of the suppression, we reconstructed them by homologous recombination. These reverse-genetic alleles rescued the viability of *rab1^ts* at 37 °C to a similar extent as *su1rab1^ts* and *su85rab1^ts* (*Figure 1C*). *uso1^G540S* was the strongest suppressor, such that *rab1^ts uso1^G540S* double mutants grew nearly as the wt at 37 °C. Nevertheless, the two alleles showed additivity, and a triple mutant carrying *uso1^E6K*, *uso1^G540S*, and *rab1^ts* grew at 42 °C, unlike either single mutant (*Figure 1C*). These data, together with the genetic mapping above, established that *uso1^E6K* and *uso1^G540S* are responsible for the suppression phenotype.

RAB1 recruits Uso1/p115 to COPII vesicles and early Golgi cisternae (*Allan et al., 2000*). Therefore, *uso1^E6K/G540S* might, by increasing the affinity of Uso1 for RAB1, compensate for the reduction in the amount of the GTPase resulting from *rab1^ts*. However, *Figure 1D* shows that both *uso1^E6K* and *uso1^G540S* rescue the lethality resulting from the complete ablation of *rab1Δ* at 30 °C, with the strongest *uso1^G540S* suppressor rescuing viability even at 37 °C, and the double mutant rescuing *rab1Δ* even at 42 °C (*Figure 1D*). In contrast, *uso1^E6K/G540S* did not rescue the lethality resulting from *arf1Δ*, nor from *sed5Δ* or *sly1Δ*, the syntaxin and the SM protein which are crucial for the formation of the ER/Golgi

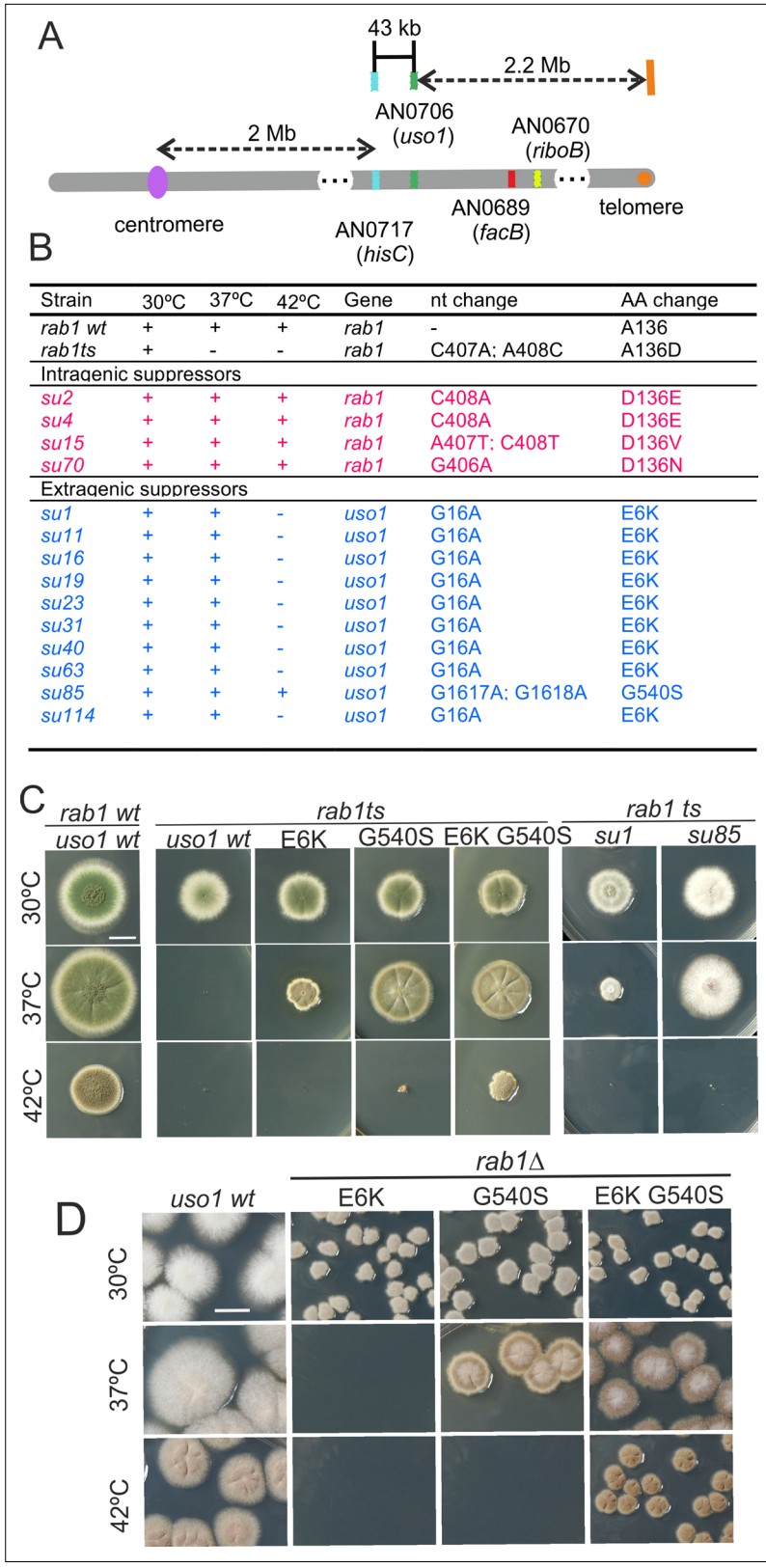

**Figure 1.** Characterization of mutations bypassing the essential role of RAB1. (**A**). Genetic map in the region surrounding *uso1* with genetic markers used as landmarks for mapping. (**B**). Molecular identification of the nucleotide changes in *suArab1ts* strains. (**C**) and (**D**): growth tests showing *rab1ts*- and *rab1Δ*-rescuing phenotypes, respectively, of individual mutations, and synthetic positive interaction between E6K and G540S. Strains produce

*Figure 1 continued on next page*

*Figure 1 continued*

either green or white conidiospores (conidiospore colors are used as genetic markers). In (**C**), strains were point-inoculated. In (**D**) conidiospores were spread on agar plates to give individual colonies.

The online version of this article includes the following source data and figure supplement(s) for figure 1:

**Source data 1.**

**Figure supplement 1.** E6K/G540S do not rescue lethality resulting from *arf1Δ*, *sly1Δ*, or *sed5Δ*.

SNARE bundle (*Figure 1—figure supplement 1*), indicating that Uso1 plays a role acting downstream of RAB1 and upstream of or in conjunction with the SNARE machinery. This role is essential for survival (*Figure 1—figure supplement 1*).

## E6K affects a previously undetected N-terminal helix, whereas G540S is located in a loop near the end of the armadillo domain

1103-residue *A. nidulans* Uso1 is similar in size to 961-residue p115 (bovine) and notably shorter than *S. cerevisiae* Uso1p (1790 residues) (*Yamakawa et al., 1996*). Thus far, atomic structures of Uso1/p115 are limited to the 600–700 residue GHD, which consists of a highly conserved α-catenin-like armadillo-fold (*An et al., 2009*; *Heo et al., 2020*; *Striegl et al., 2009*). In silico analyses robustly predict that the approximately C-terminal half of Uso1/p115 consists of a coiled-coil that mediates tethering and dimerization, but this region has not been characterized beyond low-resolution EM studies (*Yamakawa et al., 1996*). Neither crystal structures nor predictions provided information about the N-terminal extension in which Glu6Lys lies.

Thus, we used AlphaFold2, imposing the condition that the protein is a dimer (see below). *Figure 2A* shows a model with the highest confidence scores (see *Figure 2—figure supplement 1*). Like their relatives, Uso1 from *A. nidulans* contains an N-terminal GHD including a previously unnoticed short α-helix in which Glu6 is affected by E6K lies. This N-terminal extension is followed by ~34 α-helices arranged into 12 tandem repetitions of armadillo repeats (ARM1-ARM12; residues 17 through 564), each containing three right-handed α-helices except for the first two repeats. Altogether, the armadillo repeats resemble the shape of a jai alai basket. Downstream of the GHD, AlphaFold2 predicts a long extended CC between residues 674 through 1082, which would mediate dimerization (see below) (*Figure 2A*). The CC ends at a conserved CTR (*Figure 2A and D*), which includes an also C-terminal segment rich in acidic residues. In Uso1, twelve out of the last seventeen amino acids are Asp/Glu (*Figure 2A and D*). The CC and CTR regions will be collectively denoted the coiled-coil domain (CCD). Readers should note that the model of the CCD is highly speculative (see *Figure 2—figure supplement 1*), and that it is displayed with the sole purpose of visually depicting the relative size of this region compared to the globular domain, which may help to consider the potential effects of this extended region in the sedimentation data discussed below.

Even though the GHD contains Glu6 and Gly540 in the N-terminal helix and at the beginning of armadillo α-helix 29, respectively, intramolecular or intermolecular (in the context of a homodimer, see below) distances between these residues are long, arguing against the possibility that they bind a common target as components of the same interaction surface (*Figure 2B*). Indeed, the synthetic positive effect of the mutations would be consistent with their rescuing viability through different mechanisms. The previously unnoticed short α-helix predicted by AlphaFold between Phe2 and Lys12 is amphipathic (*Figure 2C*). Glu6 lies on the polar side of this helix, such that Glu6Lys increases its overall positive charge (three of the four polar residues are Lys or Arg).

## Coiled-coil mediated dimerization of Uso1: the globular head as isolated from bacteria is monomeric

It has been proposed that p115 alternates between closed and open conformations to hide or expose a RAB1 binding site present in the coiled-coil region (*Beard et al., 2005*). This would be mediated by intramolecular interactions between the globular domain and the C-terminal acidic region, which would be disrupted by the competitive binding of golgins GM130 and giantin to the latter. We addressed whether E6K/G540S promotes a conformational change in Uso1, or, alternatively, a change in the oligomerization status of the protein, by analytical ultracentrifugation. We designed seven constructs carrying a C-terminal His tag (*Figure 3*). Two corresponded to the full-length protein with

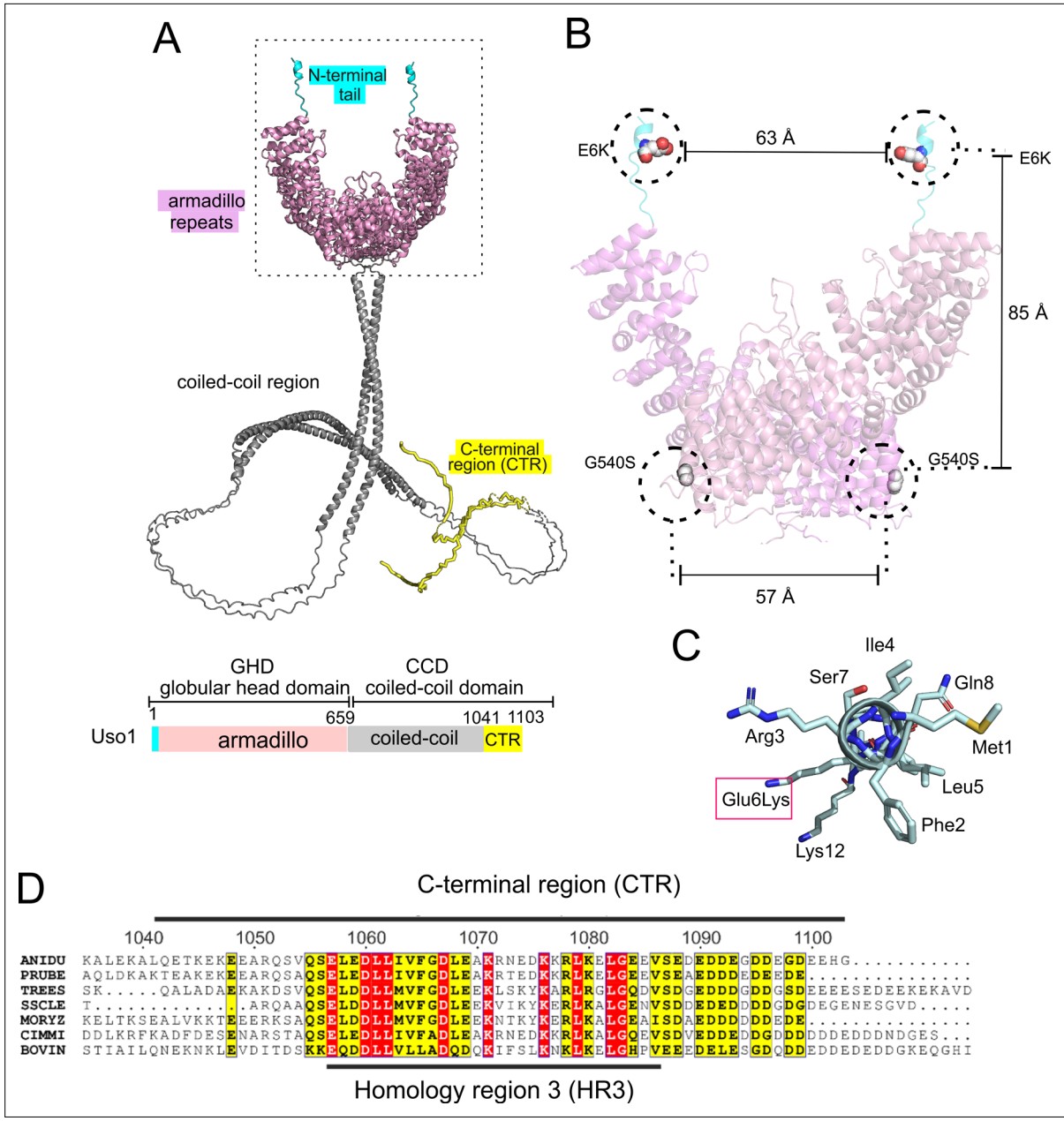

**Figure 2.** Localization of the amino acid substitutions within the Uso1 AlphaFold2 structure. (**A**). AlphaFold2 cartoon representations of *A. nidulans* Uso1 dimer. Note that the depiction of the coiled-coil domain (CCD) is highly speculative. It is included with the sole purpose of visually appreciating the relative sizes of the globular-head-domain (GHD) and the CCD domains. Confidence estimations for Uso1 models are detailed in *Figure 2—figure supplement 1*. Colors are as in the scheme below: Cyan, N-terminal tail; pink, globular head domain; gray, coiled-coil; yellow, C-terminal region (CTR). (**B**). Position of the Gly6Lys and Gly540Ser substitutions. Only the GHD of dimeric full-length Uso1 is shown. Distances between mutated residues are displayed in angstroms. (**C**). The N-terminal amphipathic α-helix affected by the Glu6Lys substitution. (**D**). Amino acid alignment of fungal sequences with mammalian p115 showing strong conservation within the CTR: ANIDU, *Aspergillus nidulans*; PRUBE, *Penicillium rubens*; TREES, *Thrichoderma ressei*; SSCLE, *Sclerotinia scleriotorum*; MORYZ, *Magnaporthe oryzae*; CIMM, *Coccidioides immitis*; BOVIN, *Bos taurus*.

The online version of this article includes the following figure supplement(s) for figure 2:

**Figure supplement 1.** AlphaFold2 predictions of Uso1.

or without E6K and G540S substitutions. The second pair included wild-type and doubly-substituted versions of C-terminally truncated Uso1 lacking the CTR (Uso1ΔCTR and Uso1$^{E6K/G540S}$ΔCTR). The third corresponded to wild-type and doubly-substituted versions of the globular domain, denoted Uso1 GHD and Uso1 GHD$^{E6K/G540S}$. The seventh construct corresponded to the Uso1 CCD. All seven proteins

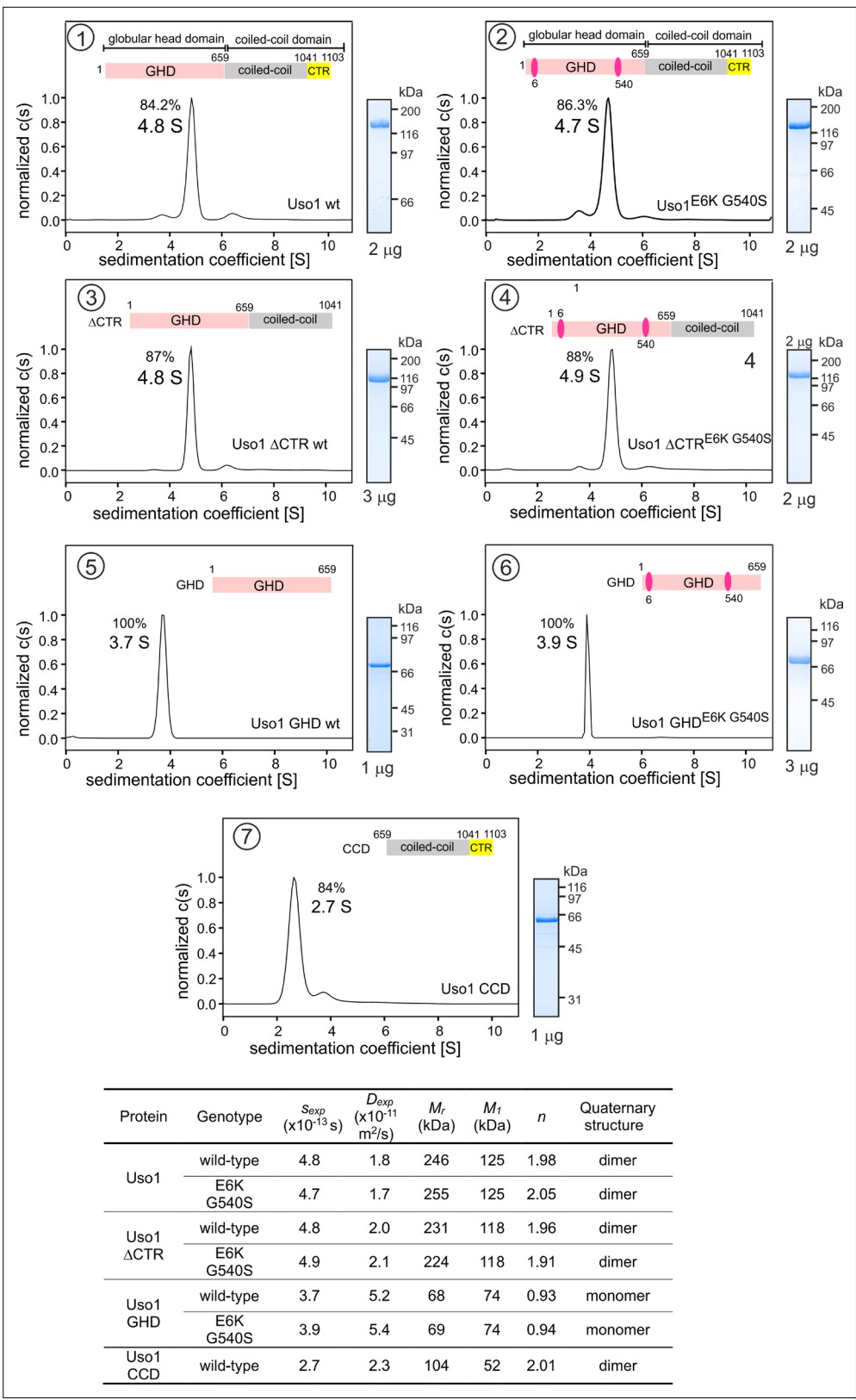

**Figure 3.** Determining molecular masses and oligomerization status of the different Uso1 constructs by velocity sedimentation analysis. The different panels display the sedimentation profiles of the protein being analyzed, with % of the main species, scheme of the corresponding constructs (color matching those in *Figure 2*), and pictures of Coomassie stained gels showing the purity of the protein preparations. The bottom table depicts the biophysical

*Figure 3 continued*

parameters of the constructs used to obtain relative molecular masses. $s_{exp}$ is the experimentally determined Svedberg coefficient; $D_{exp}$, translational diffusion coefficient of the main species; $Mr$, molecular mass deduced from Svedberg equation; M1 predicted molecular mass of the monomer; n = ($Mr$/M1).

The online version of this article includes the following source data and figure supplement(s) for figure 3:

**Source data 1.**

**Figure supplement 1.** Globular-head-domain (GHD) is a monomer across a range of concentrations.

were expressed in bacteria, purified by Ni2+-affinity and size-exclusion chromatography, and analyzed by sedimentation velocity ultracentrifugation. These experiments revealed that all protein preparations were essentially homogeneous, and thus they were used to determine the corresponding Svedberg coefficients. In addition, by dynamic light scattering we determined the translational diffusion coefficients of the constructs. With these values, we deduced the molecular mass of the different proteins using Svedberg's equation.

Wild-type and E6K/G540S full-length Uso1s showed the same sedimentation coefficients, demonstrating that the mutations do not induce a large conformational change that would have been reflected in changes in *s*-values due to differences in frictional forces. Molecular masses deduced from the Svedberg equation indicated that these full-length proteins are homodimers (*Figure 3*), in agreement with previous literature. Ablation of the conserved CTR did not result in any significant change in the sedimentation coefficient (*Figure 3, panels 3 and 4 vs. 1 and 2*), irrespective of the presence or absence of the substitutions, negating a hypothetical model in which the CTR would interact with the GHD to maintain a closed conformation (*Beard et al., 2005*). In addition, the molecular masses of the ΔCTR proteins correspond to a dimer, implying that the acidic region is not involved in dimerization either.

Notably the GHD, whether wild-type or mutant, behaved as a monomer (*Figure 3, panels 5 and 6*), which has important implications described below. In contrast, the CCD, with a predicted molecular mass of 52 kDa, behaved as a dimer of *ca*. 100 kDa (*Figure 3, panel 7*). The sedimentation coefficient of the CCD is markedly slower than that of the 70 kDa monomeric GHD, suggesting an elongated shape. These observations, together with the dimeric nature of the construct lacking the CTR, showed that dimerization is mediated by the CCD. The absence of 443 residues corresponding to the CCD plus CTR domains in the GHD construct and the monomeric nature of the latter compared to the full-length Uso1 dimer did not result in a commensurate decrease in sedimentation coefficient, which changed from 4.8 S to 3.7 S in the wild-type (note that the change in $Mr$ goes from 246 kDa in full-length Uso1 to only 68 kDa of the GHD) (*Figure 3*). These data are in line with AlphaFold2 predictions depicting Uso1 as a dimer with a globular head and an extended coiled-coil that would retard sedimentation of the protein markedly.

As with full-length Uso1, the double substitution did not alter the sedimentation coefficient of the GHD (*Figure 3, panels 5 and 6*). To buttress the conclusion that bacterially-expressed GHD is a monomer irrespective of the presence or absence of the mutations, we performed sedimentation velocity experiments using different protein concentrations ranging from 0.5 to 5 µM (*Figure 3—figure supplement 1A, B*). In all cases, the GHD behaved as a monomer. Sedimentation profiles of Uso1 GHD lacking the His-tag showed a similar behavior, establishing that the monomeric state of the mutant is not due to the tag at the C-terminal position hindering dimerization (*Figure 3—figure supplement 1C*). Therefore, sedimentation experiments did not detect any change in tertiary or quaternary structures between wild-type and mutant GHD, which is important for the interpretation of genetic data that will be discussed below.

In summary, (i) Uso1 is a dimer; (ii) The C-terminal acidic region is dispensable for dimerization and does not mediate an equilibrium between closed and open conformations; (iii) The bacterially-expressed globular domain of Uso1 is a monomer; (iv) The coiled-coil domain of Uso1 is a dimer; (v) the double E6K G540S substitution does not promote any large conformational shift in Uso1, nor does it result in a change in the oligomerization state of the protein.

## Uso1-GFP localizes to the early Golgi in a RAB1-dependent manner

The membranous compartments of the Golgi are not generally stacked in fungi, permitting the resolution of cisternae, which appear as punctate structures in different steps of maturation, by wide-field

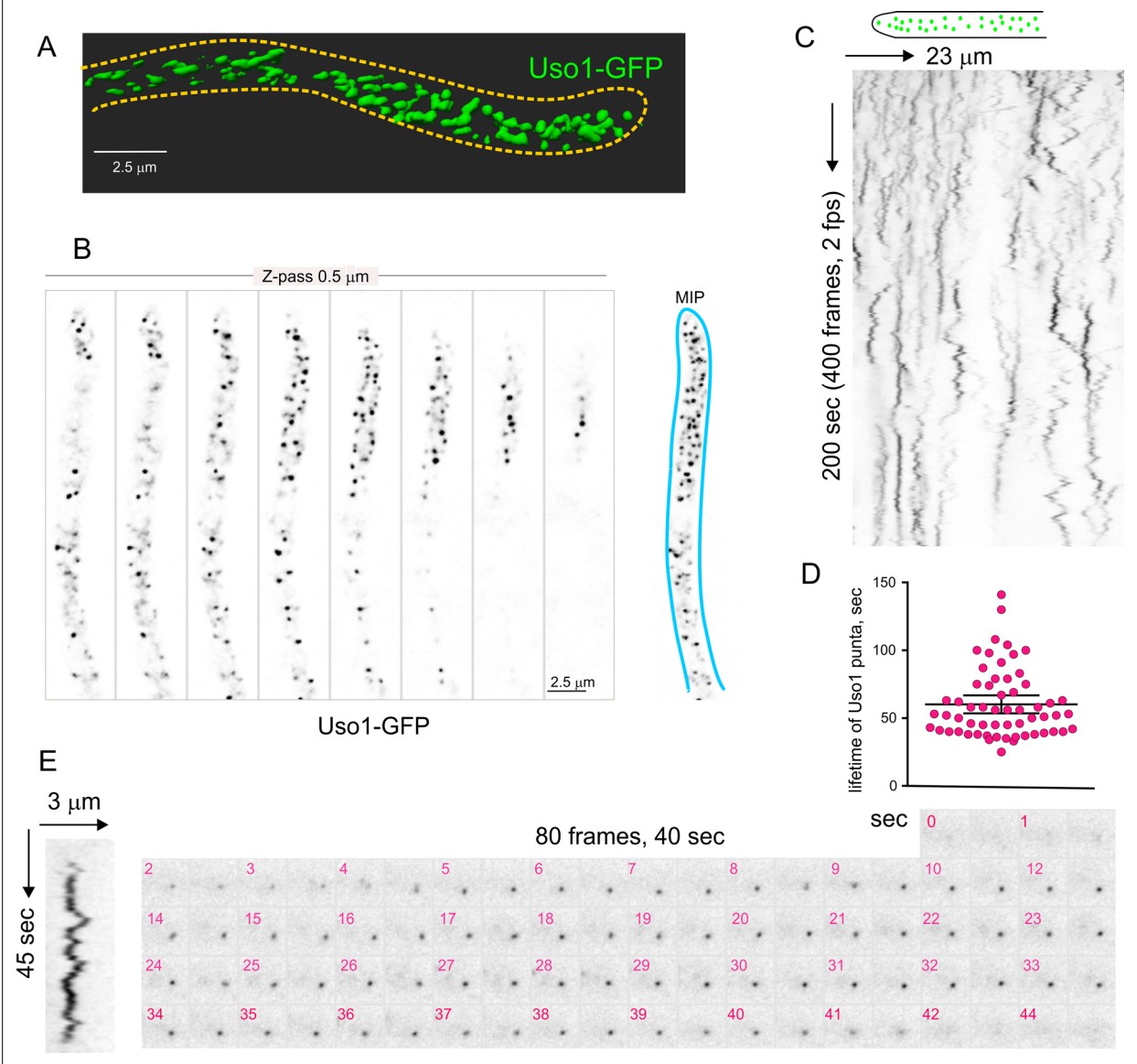

**Figure 4.** Subcellular localization of Uso1. (**A**). Uso1-GFP localizing to punctate cytoplasmic structures, 3D shaded by software. (**B**). Sections of a deconvolved Z-stack and its corresponding maximal intensity projection (MIP). Uso1-GFP in inverted grayscale for clarity. (**C**). Kymograph showing the transient recruitment of Uso1 to punctate cytoplasmic structures. (**D**). Average time of residence of Uso1 in these structures. Error bars, 95% CI. (**E**). Example of one such structure visualized with a kymograph and with the corresponding movie frames (*Video 4*).

The online version of this article includes the following figure supplement(s) for figure 4:

**Figure supplement 1.** Methodology for tracking the half-life of Uso1-GFP on punctate structures.

fluorescence microscopy (*Losev et al., 2006*; *Matsuura-Tokita et al., 2006*; *Pantazopoulou and Peñalva, 2011*; *Pinar et al., 2013*; *Wooding and Pelham, 1998*). While Uso1 is predicted to localize to the Golgi, studies of its localization in fungi are limited (*Cruz-Garcia et al., 2014*; *Sánchez-León et al., 2015*). Therefore, we tagged the *A. nidulans uso1* gene endogenously with GFP.

*Figure 4A* and *Video 1* depicting a software-shadowed 3D reconstruction of a Uso1-GFP hypha, as well as consecutive sections of deconvolved z-stacks in *Figure 4B* show that Uso1-GFP localizes to puncta polarized towards the tip, often undergoing short-distance movements (see *Figure 4C* and *Figure 4—figure supplement 1*). These puncta are smaller and more abundant than those reported for other markers of the Golgi, which suggested that they might represent domains rather than complete cisternae. Notably, 3D (x, y, t) movies revealed that Uso1 puncta are transient, recurrently appearing, and disappearing with time (*Figure 4C*). That this recurrence did not reflect that

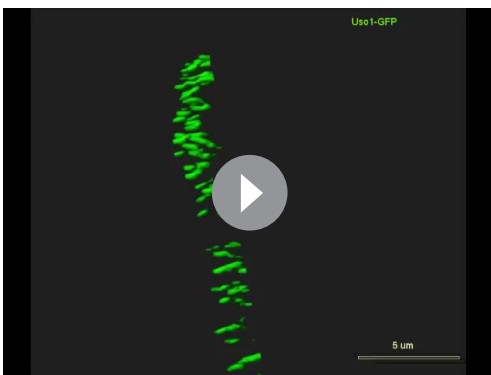

**Video 1.** Shaded 3D reconstruction of a hypha expressing Uso1-GFP.
https://elifesciences.org/articles/85079/figures#video1

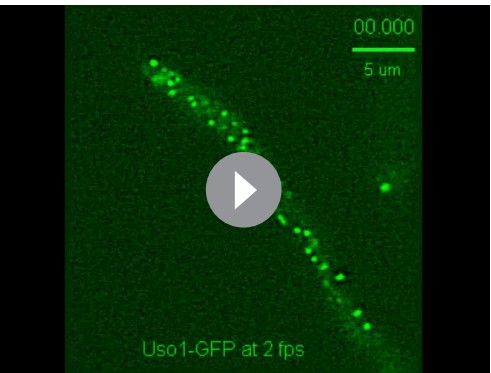

**Video 3.** Dynamics of Uso1-GFP at 2 fps 3D acquisition (200 frames) showing the dynamics of Uso1-GFP. Time resolution, 2 fps.
https://elifesciences.org/articles/85079/figures#video3

the puncta go in-and-out of focus was established with 4D (x, y, z, t) movies, which revealed a similar behavior of Uso1 irrespective of whether 3D or 4D microscopy was used (*Video 2*). Therefore, we constructed movies with middle planes only (i.e. 3D x, y, t series). After adjustment of live imaging conditions, we achieved a 2 fps time resolution with relatively low bleaching for time series consisting of 400 photograms (*Video 3*). These conditions sufficed to track Uso1 puncta over time using kymographs traced across linear ROIs covering the complete width of the hyphae (*Figure 4C*). However, as the abundance of Uso1 puncta made automated analysis of Uso1 maturation events troublesome, we tracked them manually with the aid of 3D (**x, y, t**) representations generated with Imaris software combined with direct observation of photograms in movies (*Figure 4C* and *Figure 4—figure supplement 1*). The boxed event magnified in *Figure 4E* (see *Video 4*) illustrates a prototypical example. The right *Figure 4E* montage shows frames corresponding to this event for comparison. We analyzed n=60 events, which gave an estimation of the average half-life of Uso1 residing in puncta of 60 sec+/−25 S.D. (*Figure 4D*).

Nakano and co-workers have proposed that the transfer of lipids and proteins between ER exit sites (ERES) and the early Golgi occurs through a kiss-and-run mechanism (*Kurokawa et al., 2014*). Because Uso1-GFP punctate structures resemble, in size and abundance, ER exit sites labeled with COPII components, we studied Uso1-GFP cells co-expressing Sec13 endogenously labeled with mCherry (*Bravo-Plaza et al., 2019*)(*Hernández-González et al., 2019*). The maximal intensity projection (MIP) shown in *Figure 5A*, and *Video 5* show that the two markers are closely associated, but only in a few instances they showed colocalization. These examples did not represent simple overlap, as they were found to colocalize in the Z dimension using orthogonal views or montages (*Figure 5B and C*). These observations have not been pursued further with time-resolved sequences, but at the very least we can conclude that the reporters are

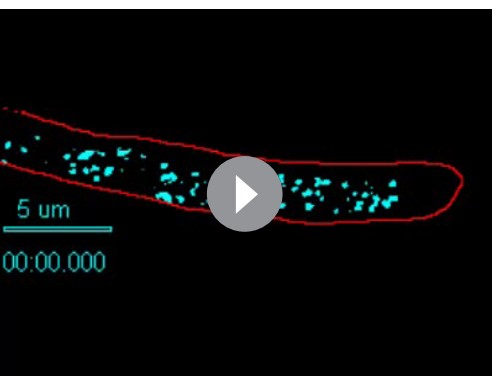

**Video 2.** 4D acquisition showing the dynamics of Uso1-GFP. 4D (x, y, z, t) in which Z-stacks were acquired at a rate of 1 frame every 2.6 s.
https://elifesciences.org/articles/85079/figures#video2

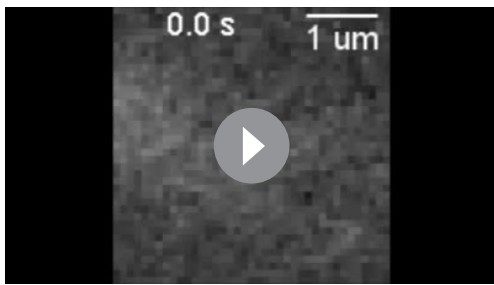

**Video 4.** Single Uso1-GFP cisterna tracked over time Example of Uso1-GFP cisterna. The video contains 96 photograms acquired at 2fps.
https://elifesciences.org/articles/85079/figures#video4

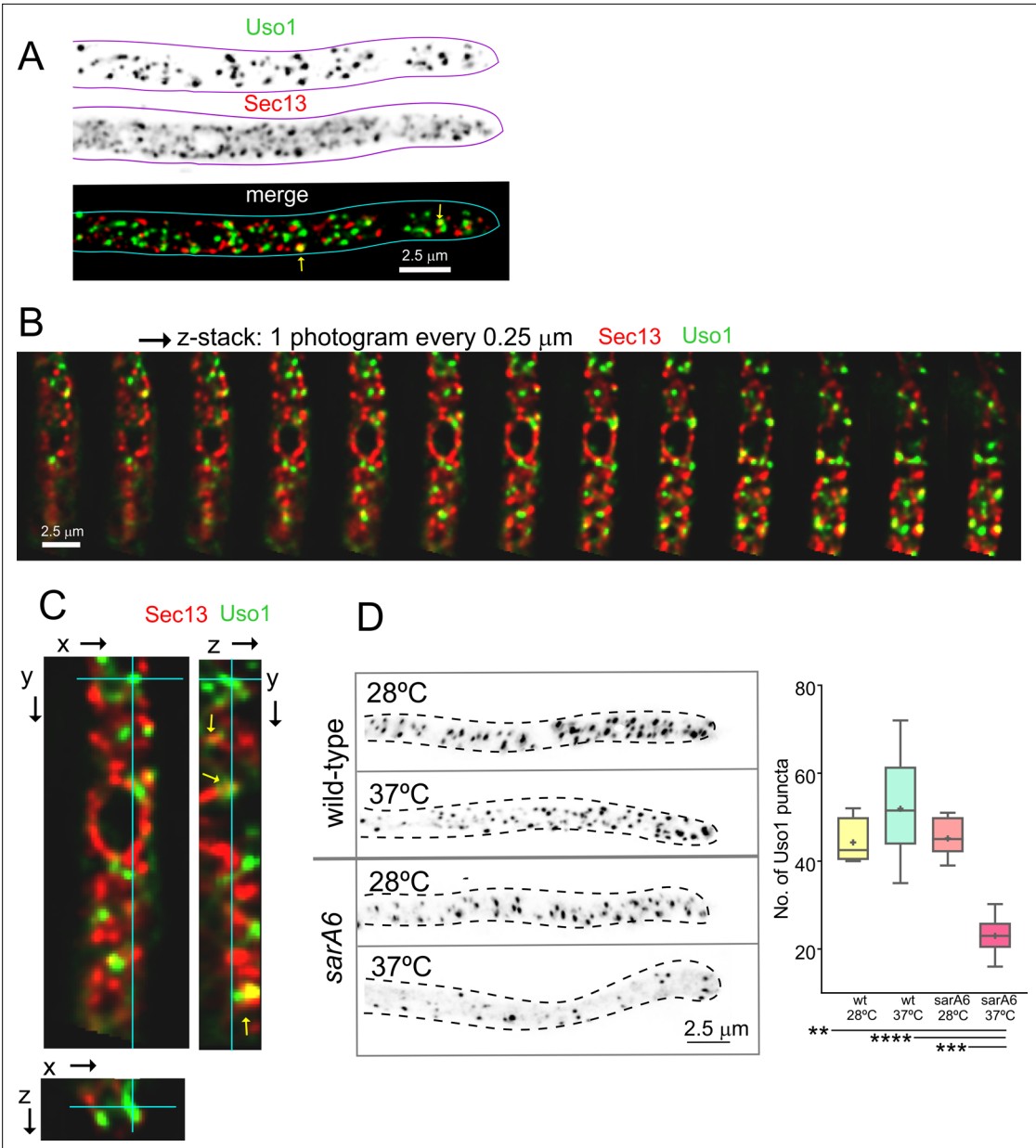

**Figure 5.** Uso1 puncta do not colocalize with ER exit sites (ERESs). (**A**). Low extent co-localization of Sec13 ERES and Uso1 structures. Z-stacks for the two channels were acquired simultaneously, deconvolved, and represented as maximal intensity projections (MIPs). Two rare examples of colocalization are arrowed. (**B**). Photograms of a dual channel Z-stack with a Sec13-labeled nuclear envelope focused in the middle plane, illustrating that while some puncta show colocalization, the red Sec13 signal, and the green Uso1 signal do not usually overlap. (**C**). A MIP of the same z-stack showing orthogonal views with some overlapping puncta (arrows). (**D**). A ts mutation in the *sarA* gene encoding the SarA$^{Sar1}$ GTPase governing ER exit markedly reduces the number of Uso1-GFP puncta upon shifting cells to restrictive conditions. Box-and-whisker plots: Statistical comparison was made using one-way ANOVA with Dunn's test for multiple comparisons. Whiskers are in Tukey's style: Only significant differences were indicated, using asterisks.

closely associated in space. Next, we investigated, using *sarA6*, a temperature-sensitive allele of the gene encoding *A. nidulans* SAR1 (*Hernández-González et al., 2015*), whether the punctate Uso1 structures are dependent on this master GTPase regulating COPII biogenesis. *Figure 5D* shows that this is indeed the case; the number of Uso1-GFP puncta was significantly reduced relative to the wt when cells were shifted from 28 to 37°C, indicating that Uso1 populates a membrane compartment subordinated to Sar1, arguably one with early Golgi identity.

Therefore, we filmed Uso1-GFP along with different Golgi markers (*Figure 6*). Uso1-GFP showed no overlap (Pearson's coefficient 0.17 ± 0.06 S.D., n=16 cells) with cisternae labeled with mCherry-Sec7,

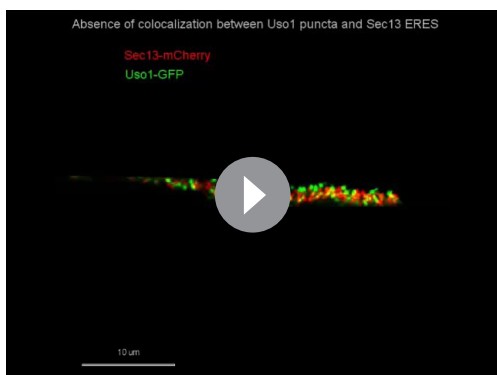

**Video 5.** 3D reconstruction of a hypha expressing fluorescently labeled Uso1-GFP and Sec13-mCh. There is little colocalization between Uso1-GFP and Sec13 ERES.

https://elifesciences.org/articles/85079/figures#video5

the late Golgi ARF1 GEF that is a prototypic marker of the TGN (*Arst et al., 2014*; *Day et al., 2018*; *Galindo et al., 2016*; *Halaby and Fromme, 2018*; *Losev et al., 2006*; *McDonald and Fromme, 2014*; *Pantazopoulou, 2016*; *Pantazopoulou and Glick, 2019*; *Richardson et al., 2012*; *Richardson et al., 2016*) (*Figure 6A–C*; *Video 6*). In contrast, visual observation of cells expressing mCh-Sed5 and Uso1-GFP revealed a substantial, yet incomplete, overlap of the reporters (*Figure 6A and B*), reflected in a Pearson's coefficient of 0.44 ± 0.04 S.D., n=15 cells (*Figure 6C*). The Qa syntaxin Sed5 drives the fusion of COPII vesicles with early Golgi cisternae, with Qb, Qc and R-SNAREs Bet1, Bos1, and Sec22, and mediates intra-Golgi trafficking, with Qb, Qc and R-SNARES Sft1, Gos1, and Ykt6, respectively (*Banfield et al., 1995*; *McNew et al., 2000*; *Parlati et al., 2002*; *Pelham, 1999*; *Wooding and Pelham, 1998*). These data suggest that Uso1 localizes to a subset of early Golgi cisternae/membranes containing Sed5. GeaA[Gea1,2] is the only *A. nidulans* homolog of the *S. cerevisiae* early Golgi ARF1 GEFs Gea1 and Gea2 dwelling at the early Golgi (*Arst et al., 2014*; *Gustafson and Fromme, 2017*; *Muccini et al., 2022*; *Pantazopoulou, 2016*; *Park et al., 2005*; *Wright et al., 2014*). Overlapping of Uso1 with GeaA[Gea1,2] was more conspicuous than with Sed5 (*Figure 6A–C*), which was reflected in an increased Pearson's coefficient to 0.52 ± 0.06 S.D., n=16 cells. Of note, mammalian GeaA (GBF1) and Uso1 (p115) interact (*Garcia-Mata and Sztul, 2003*). Taken together, these data suggested that Uso1 localizes to Golgi cisternae in the early stages of maturation.

In both mammalian cells and in yeasts Uso1/p115 has been shown to be recruited to early Golgi membranes by RAB1, which is activated by the TRAPPIII GEF on COPII vesicles after they bud from the ER (*Allan et al., 2000*; *Cai et al., 2007*; *Lord et al., 2011*; *Yuan et al., 2017*). Therefore, we imaged Uso1 and RAB1, which revealed that indeed Uso1 colocalized with RAB1 (Pearson's 0.61 ± 0.07 S.D., n=20 cells) (*Figure 6A–C*). Altogether, the above microscopy data strongly suggest that Uso1 is transiently recruited to early Golgi membrane domains enriched in RAB1, agreeing with the accepted view that RAB1 acts by recruiting Uso1.

RAB1 plays roles extending beyond the Golgi (*McDonald and Fromme, 2014*), which very likely explains why its colocalization with Uso1 is not complete. To test this interpretation, we used Rer1, which is a cargo receptor that continuously cycles between the ER and the Golgi, localizing in the steady state to the early Golgi (*Pinar et al., 2013*; *Sato et al., 2003*). *Figure 6D and E* shows that Rer1 and Uso1 colocalize almost completely which is translated into a Pearson's coefficient of 0.81+/−0.03 95% C.I. Therefore, this localization of Uso1 is consistent with it playing a role at the entry of the Golgi.

## Uso1 delocalization and partial Golgi disorganization after RAB1 impairment rescued by E6K G540S

We next tested if the subcellular localization of Uso1 is dependent on RAB1, and if this dependency can be bypassed by E6K/G540S. To this end we first showed that in a *RAB1*[+]background endogenously tagged wild-type and E6K/G540S Uso1-GFP have the same punctate localization pattern (*Figure 7A*), and that both supported vigorous wt growth (*Figure 7B*, rows 1, 3, 5, and 7), indicating that the tagged proteins are functional. Next, we introduced in these strains *rab1*[ts] (*Jedd et al., 1995*; *Pinar et al., 2013*) by crossing. This allele, which completely prevents growth at 37 °C, is a hypomorph at permissive (25–30°C) temperatures, which permitted testing RAB1 dependence under standard microscopy conditions (28 °C). *Figure 7A* shows that wt Uso1-GFP was largely delocalized to the cytosol by *rab1*[ts], even though deconvolution revealed the presence of a few faint specks of Uso1-GFP in the cytosol that stood up against the background of cytosolic haze, consistent with the expectation that Uso1 would not be completely delocalized given that this strain, albeit sick, is able to grow at

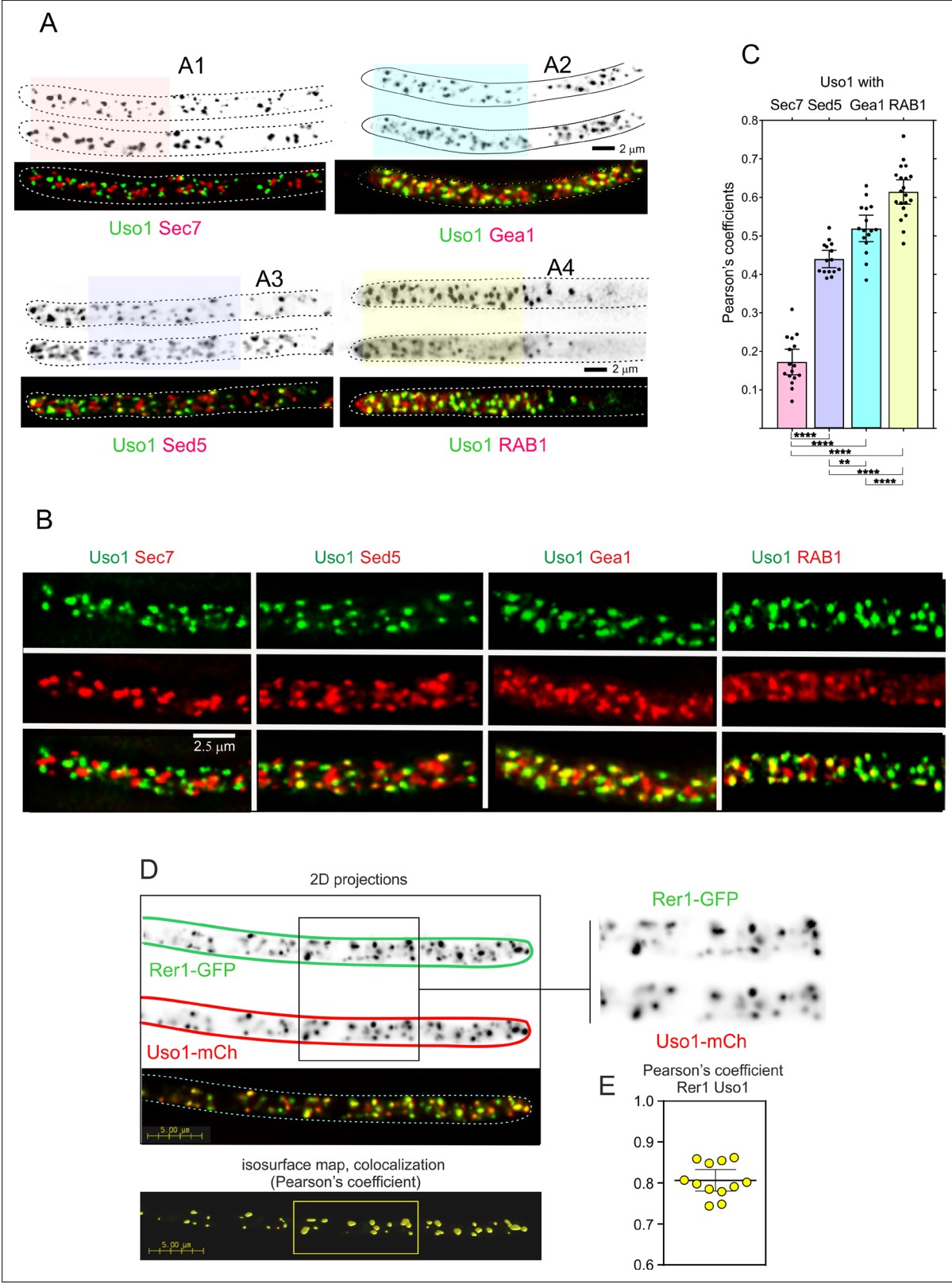

**Figure 6.** Uso1 localizes to Rer1-containing Golgi cisternae. (**A**). Tip cells showing Uso1 colocalization with the indicated subcellular markers. Images are maximal intensity projections (MIPs) of deconvolved Z-stacks. (**B**). Magnified images of the color-coded shaded regions of the cells shown in A. (**C**). Pearson's coefficients of the different combinations. (**D**) Gray scale: MIPs of deconvolved z-stacks of the individual Rer1-GFP and Uso1-mCh channels, with their merge at the bottom. The boxed section of the channels is magnified on the right. The bottom shows a yellow isosurface map

*Figure 6 continued on next page*

*Figure 6 continued*

corresponding to the regions with high Pearson's coefficient (constructed with Huygens software). (**E**) Pearson's coefficient in n=12 cells. Error bars indicate 95% CI

this temperature (*Figure 7B*, row 4) The normal, punctate localization of Uso1 was restored by the E6K/G540 substitutions, establishing that Uso1 localization to membranes is subordinated to RAB1. Notably, the wt punctate pattern of Uso1 localization was restored by the Uso1 E6K/G540S mutant substitution, correlating with the correction of the synthetic growth defect (*Figure 7B*, compare rows 2, 4, 6, and 8). All these data indicated that the localization of Uso1 is compromised when RAB1 function is impaired, and that the E6K/G540S substitutions augment Uso1 affinity for a membrane anchor(s) independent of RAB1. They additionally suggest that the principal physiological role of RAB1 is ensuring the proper localization of Uso1.

Once the effect that the double E6K/G540S substitution has on Uso1-GFP subcellular distribution had been established, we addressed whether its mis-localization had any effect on the organization of the Golgi using Gea1 as a reporter, after confirming that this combination of endogenously tagged proteins does not result in a synthetic growth defect. Correlating with redistribution of wild-type Uso1 to the cytosol in a *rab1ts* background, co-localization experiments (*Figure 7C*), showed that Gea1 cisternae were also affected, with apparently less punctate structures, which in turn co-localized with the remnants of Uso1, suggesting that they have Golgi identity. This effect was remediated if E6K/G540S Uso1-GFP rather than the wild-type, was present (*Figure 7C*). We confirmed this conclusion by counting the number of Gea1 cisternae in the apex-proximal 20 µm of hyphae. (*Figure 7D*) shows that *rab1+* cells, whether carrying or not E6K/G540S Uso1, and *rab1ts* cells carrying E6K/G540S Uso1, where indistinguishably normal in this particular regard. In sharp contrast, *rab1ts* cells carrying wild-type Uso1 contain a significantly lower number of Gea1 cisternae. Thus, we concluded that the integrity of the Golgi is affected by *rab1ts* at the permissive temperature and that this disruption is corrected by the presence of E6K/G540S Uso1.

Delocalization of Uso1 in the *rab1ts* background was not solely dependent on RAB1. Wild-type *uso1-GFP* displayed a synthetic negative interaction with *rab1ts* (*Figure 7B*, compare lanes 2 and 4 at 30 °C), suggesting that the presence of GFP in the C-terminus interferes with a RAB1-independent mechanism that facilitates its recruitment to membranes (see below).

## Genetic evidence that a network involving the CTR and the Grh1/Bug1 golgin contributes to the recruitment of Uso1 to membranes

To follow up on the above observation, we focused on golgins. In mammalian cells, the C-terminal region of p115 interacts with GM130, a golgin that is recruited to the early Golgi by GRASP65 (*Beard et al., 2005*). The equivalent proteins in budding yeast are denoted Bug1 and Grh1 (*Behnia et al., 2007*; *Figure 8A*).

GRASP65 contains two C-terminal PDZ [post synaptic density protein (PSD95), *Drosophila* disc large tumor suppressor (Dlg1), and zonula occludens-1 protein (zo-1)] domains, which bind an also C-terminal peptide in GM130 (*Hu et al., 2015*). Similar to its metazoan counterparts, *Aspergillus* Grh1 contains an N-terminal α-helix and two PDZ domains, in this case, followed by ~130 disordered residues (*Figure 8B*, *Figure 8—figure supplement 1A and C*). We modeled the Grh1-Bug1 interaction using AlphaFold2. Residues 666–675 of the C-terminal peptide of Bug1 bind to a hydrophobic cleft located between PDZ1 and PDZ2. Bug1 Leu668 and 670 coordinate their side chains with residues from both PDZ domains (e.g. Phe45 and Trp44 in PDZ1 and Trp171 and Val179 in PDZ2). A second interaction involves Bug1 C-terminal residues 683–690 fitting within a second groove in PDZ1, such

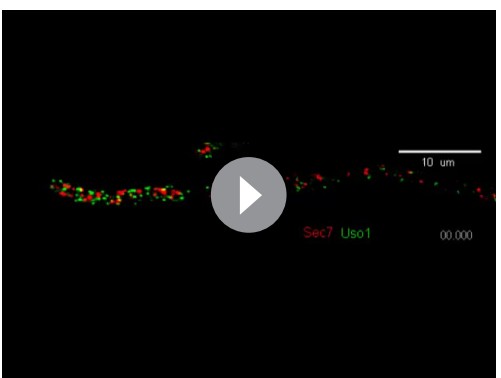

**Video 6.** 4D video (1 fpm) of a hypha expressing fluorescently labeled Uso1-GFP and Sec7-mCh Uso1 does not colocalize at all with the TGN marker Sec7.
https://elifesciences.org/articles/85079/figures#video6

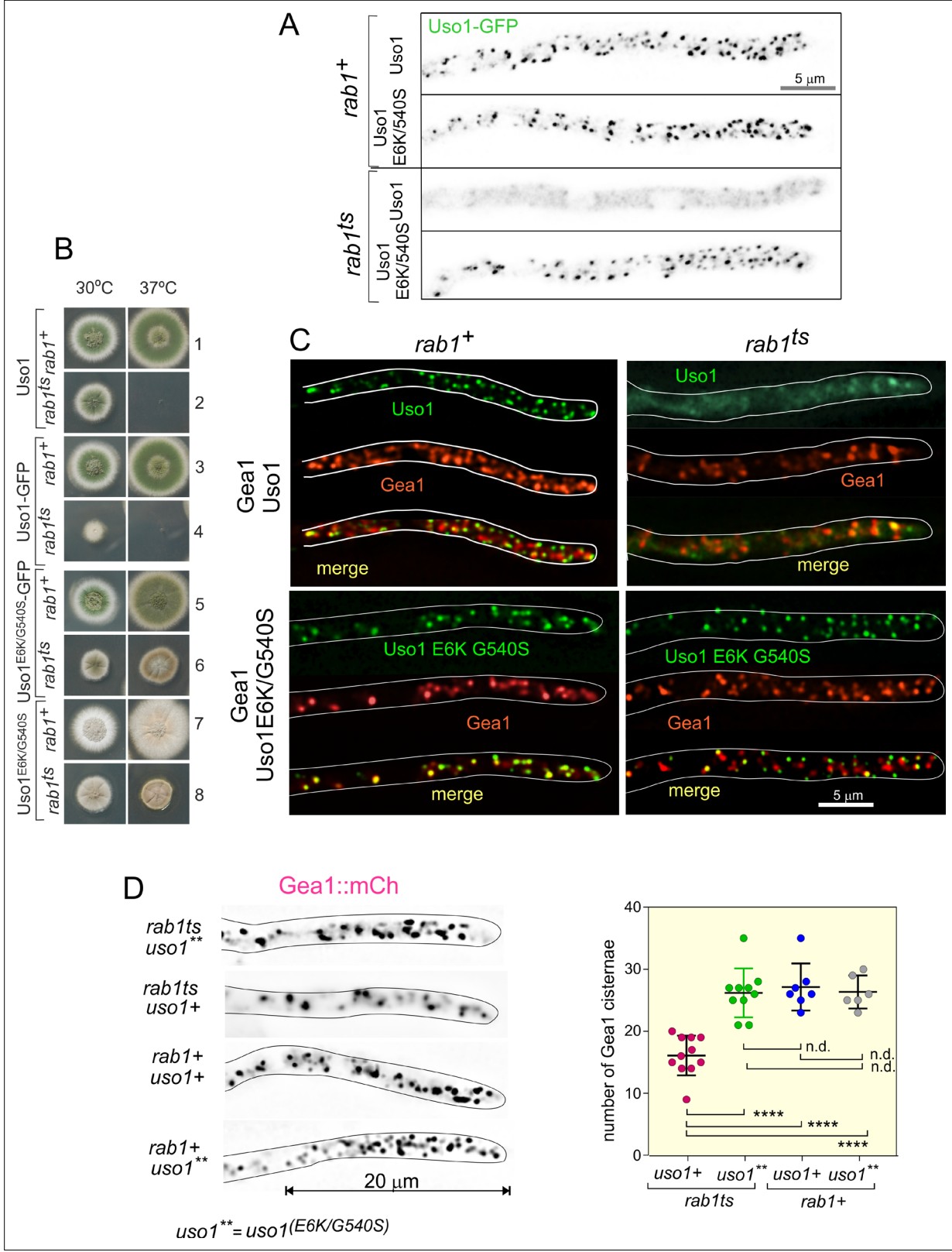

**Figure 7.** Uso1 localization to punctate structures is dependent on RAB1. (**A**). Complete de-localization of Uso1-GFP to the cytosol by *rab1^ts^* and relocalization by E6K/G540S. (**B**). Uso1-GFP and *rab1ts* show a synthetic negative interaction that is rescued by the E6K/G540S double substitution. Strains in lanes 7 and 8 carry the *wA2* mutation resulting in white conidiospores. (**C**) In *rab1^ts^* cells, punctate Uso1-GFP structures largely disappear, correlating with Golgi disorganization (GeaA/Gea1-mCh, see text). The presence of E6K/G540S relocalizes Uso1-GFP to puncta and restores Golgi

*Figure 7 continued on next page*

Figure 7 continued

organization. (**D**) Representative images (maximal intensity projections(MIPs) of deconvolved Z-stacks) of hyphae used to quantify the number of cisternae within the apex-proximal 20 μm. Datasets were compared by one-way ANOVA with Tukey's multiple comparison test. Error bars represent S.D.

that the four C-terminal residues form a β-strand extending the β-sheet of N-terminal PDZ1 domain (*Figure 8B*, *Figure 8—figure supplement 1B*). Further genetic evidence that a network of interactions similar to that acting in yeast and mammalian cells operates in the *A. nidulans* ER-Golgi interface was obtained by constructing strains with combinations of gene-replaced alleles. These consisted of *uso1ΔCTR* encoding Uso1 lacking the C-terminal region (residues 1–1041) and containing or not E6K G540S, *rab1ts*, and deletion alleles of the *Aspergillus BUG1* (AN7680) and *GRH1* (AN11248) genes (*Figure 8A*). Combining *rab1ts* with *uso1ΔCTR* resulted in a synthetic negative interaction at 30 °C akin to that seen with *uso1-GFP* (lanes 2 and 4 in *Figure 8C*). That the E6K/G540S double substitution rescued this negative interaction strongly indicates that the CTR cooperates with RAB1 in the recruitment of Uso1 to membranes (*Figure 8C*, lanes 2, 4, 6, and 8). We note that the control wild-type strain used in these experiments contains a construct completely analogous to the mutant allele, ruling out that the genetic manipulation (for example the introduction, linked to the *uso1* locus, of a selection marker, potentially chromatin-disruptive) is causative of the observed phenotype. Another trivial explanation that we ruled out by Western-blot analysis was that the deletion of the 62 C-terminal residues in *uso1ΔCTR* resulted in increased degradation, which it did not (*Figure 8D*).

If interactions involving the CTR of p115 were conserved in fungi, *BUG1* (GM130 equivalent) and *GRH1* (GRASP65) should also show a synthetic negative interaction with *rab1ts*. *Figure 8E* (lanes 1, 3, and 5) shows that neither *bug1Δ* nor *grh1Δ* affects growth. However, both deletion alleles showed a strong synthetic negative interaction with *rab1ts* (*Figure 8E*, lanes 2, 4, and 6). Remarkably, the synthetic negative phenotype was rescued by the presence of E6K/G540S substitutions in Uso1, further suggesting that they promote Uso1 recruitment to its locale of action, compensating for the loss of the Bug1-Uso1 CTR interaction. We conclude that interaction involving the CTR of Uso1 and the BUG1/GRH1 complex cooperates with RAB1-mediated mechanisms to recruit Uso1 to membranes.

## The GHD of Uso1 carrying the double E6K/G540S substitution supports cell viability

Uso1 has traditionally been considered the archetype of a coiled-coil tether recruiting ER-derived vesicles to the *cis*-Golgi. In *S. cerevisiae*, *uso1-1* (*Sapperstein et al., 1996*), is a C-terminally truncating, conditionally-lethal *ts* allele whose encoded protein still retains 20% of the coiled-coil region. In contrast, *uso1-12* and *uso1-13* removing the complete coiled-coil region are lethal (*Seog et al., 1994*). Thus, we asked whether the double E6K/540 S substitution bypasses the requirement of the coiled-coil region. To this end, we constructed, by gene replacement, a *uso1^GHD* allele expressing a protein truncated immediately after the GHD, lacking residues 660–1103. By heterokaryon rescue, we demonstrated that this allele is lethal (*Figure 9A*). Unexpectedly, given that the *rab1ts* suppressor substitutions lie outside the coiled-coil region, the equivalent allele containing E6K/G540S sufficed for the fungus to survive at 30 °C. This result has two key implications: that the coiled-coil of Uso1 is not essential for survival and as the GHD of Uso1 is a monomer in solution, it follows that dimerization of Uso1 is not essential either (see also below).

In view of these unexpected results, we wondered whether the structure of the GHD synthesized in bacteria differed from the physiological form in *Aspergillus*, such that the GHD was a dimer in vivo. To address this possibility, we ran an *Aspergillus* cell-free extract expressing HA3-tagged Uso1 GHD through a Sepharose column. As a control, we ran in parallel a sample of bacterially-expressed His-tagged GHD used in sedimentation velocity experiments. Western blot analysis of the fractions (*Figure 9B*) demonstrated that both proteins eluted at the same position, corresponding to that expected for a globular protein with the size of the GHD. As controls, we ran in the same column a similar pair of proteins corresponding to full-length Uso1. The bacterially-expressed and the native Uso1 proteins also co-eluted (*Figure 9B*), but this time at a position corresponding to a highly elongated dimer, consistent with sedimentation velocity experiments. Although we cannot rule out the possibility that the GHD dimerizes in situ upon SNARE binding, the GHD present in *Aspergillus* extracts as expressed from its own promoter is a monomer, strongly suggesting that it does not require dimerization to sustain viability if it carries the double mutant substitution that bypasses the

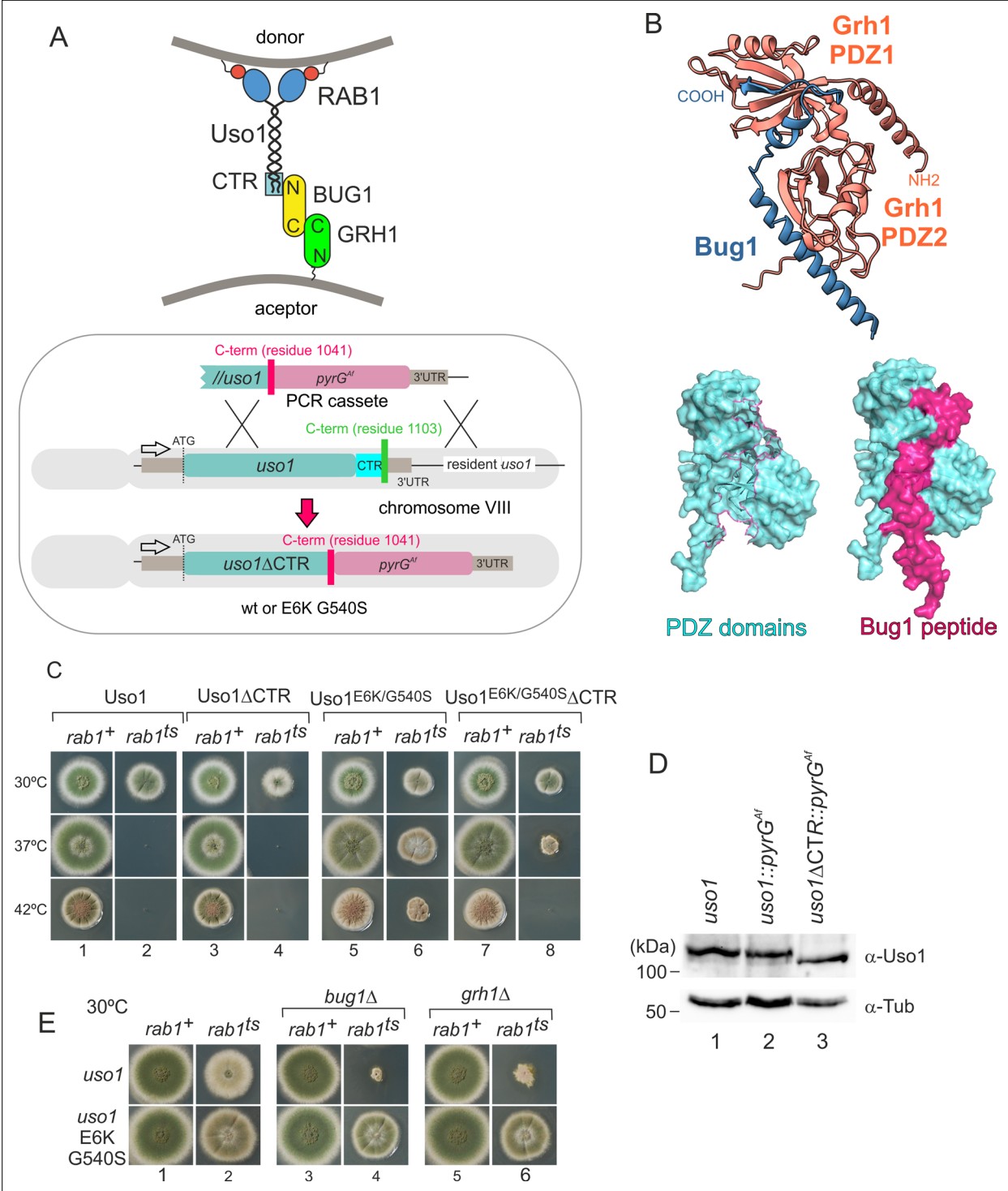

**Figure 8.** Genetic evidence shows that the C-terminal region (CTR) region of Uso1 contributes to its recruitment to membranes. (**A**). Top, scheme of the predicted interactions. Bottom, engineering a gene-replaced allele lacking the CTR domain by homologous recombination. (**B**). The Bug1 C-terminal residues fit into the groove formed between the two Grh1 PDZ domains and into the pocket of the N-terminal PDZ domain (PDZ1). (**C**). A gene-replaced *uso1ΔCTR* allele encoding a protein truncated for the CTR domain shows a synthetic negative interaction with *rab1ts*. (**D**). Western blot analysis. Removal of the CTR does not result in Uso1 instability. (**E**). *bug1Δ* and *grh1Δ* show a synthetic negative interaction with *rab1ts* that is rescued by the double E6K/G540S substitution in Uso1.

The online version of this article includes the following source data and figure supplement(s) for figure 8:

**Source data 1.** Raw images for western blots in panel D and uncropped pictures with used exposures and regions indicated.

**Figure supplement 1.** AlphaFold2 modeling of Grh1-Bug1.

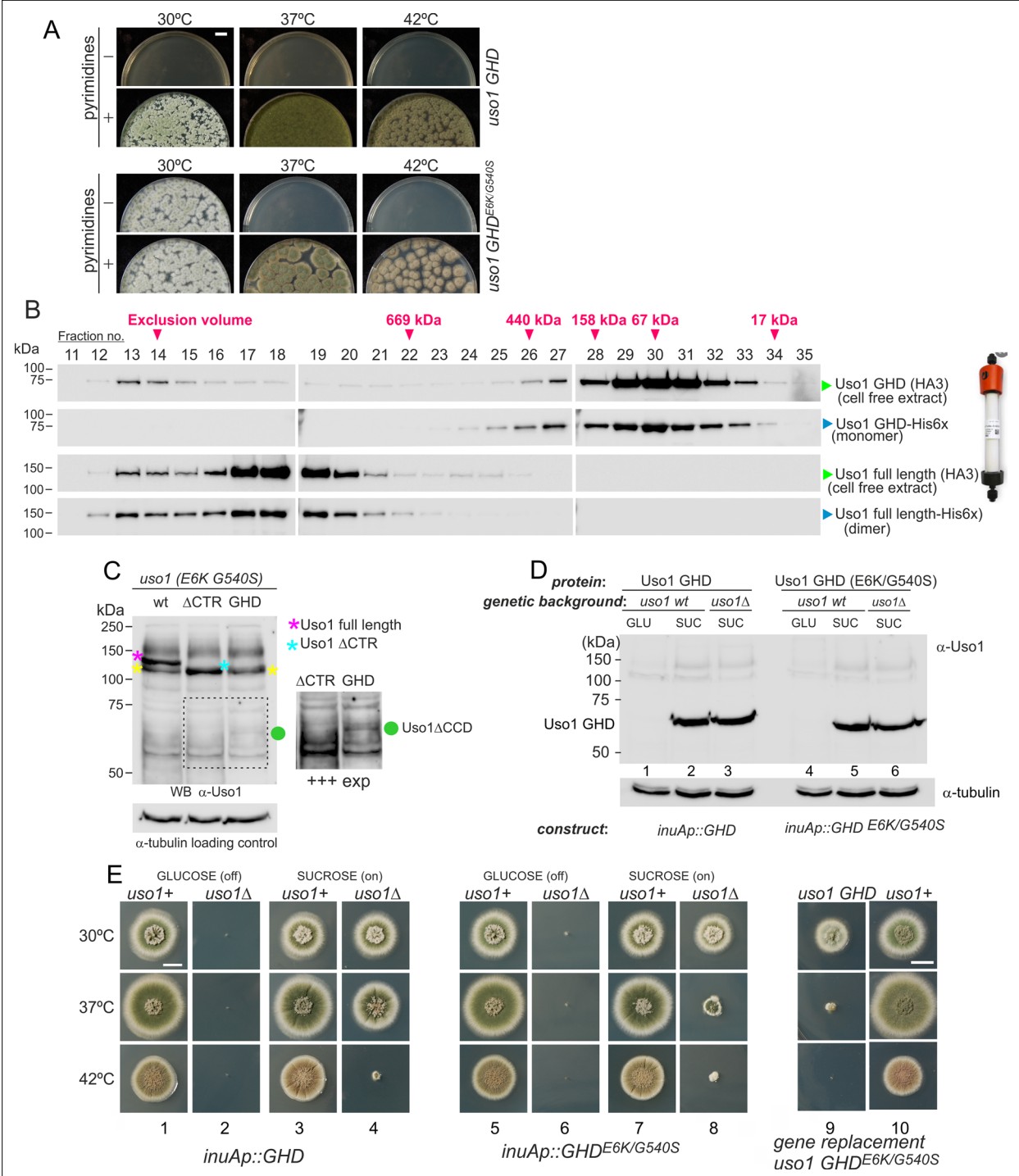

**Figure 9.** The globular-head-domain (GHD) of Uso1 is sufficient to support cell viability. (**A**). Gene-replaced *uso1*[GHD] allele carrying the double E6K/G540S substitution is sufficient to rescue viability at 30 °C, but not at higher temperatures. (**B**). The Uso1 GHD is a monomer in cell extracts. Fractions collected from Superose columns loaded with the indicated protein extracts and reference His-tagged proteins were collected and analyzed by western blotting with α-HA and α-His antibodies. (**C**) Truncating Uso1 after the GHD results in markedly reduced protein levels, as determined by α-Uso1 GHD western blotting. The band (yellow asterisks) moving slower than Uso1 (magenta asterisk) and at nearly the same position as Uso1 ΔCTR (blue asterisk) represents cross-reacting contaminants unrelated to Uso1. The right panel shows a longer exposure for the indicated region, to reveal the faint GHD band (green dot). (**D**). Overexpression of wild-type and E6K/G540S mutant GHD, under the control of the *inuA* promoter, which is turned off on glucose and induced on sucrose. Western blots reacted with α-Uso1 GHD antiserum. (**E**). Overexpressed GHD, be it E6K/G540S or wild-type, as the only source of Uso1 supports viability.

*Figure 9 continued on next page*

*Figure 9 continued*

The online version of this article includes the following source data for figure 9:

**Source data 1.** For panels B, C, D; raw images for western blots and uncropped pictures with used exposures and regions indicated.

requirement for RAB1. These data strongly argue against tethering being *the* essential physiological role that Uso1 plays.

We next investigated why the E6K/G540S GHD suffices for viability only at 30 °C. Often thermo-sensitivity results from protein instability, which is enhanced at high temperatures. Western blot analysis of the allele-replaced strain expressing E6K G540S GHD as the only source of Uso1 revealed that levels of the truncated Uso1 mutant were minuscule relative to the wt or to the equivalent *ΔCTR* allele (*Figure 9C*). We reasoned that increasing expression would result in E6K/G540S GHD supporting growth over a wider range of temperatures. Thus, we drove its expression with the promoter of the inulinase *inuA* gene, which is inducible by the presence of sucrose in the medium and almost completely shut off on glucose (*Hernández-González et al., 2018*; *Peñalva et al., 2020*). Initially, we tested wild-type and E6K/G540S GHD in a *uso1+* background. This had no phenotypic consequences despite the fact that western blots confirmed that the truncated proteins were being overexpressed (*Figure 9D*; *Figure 9E*, lanes 1, 3, 5, 7, and 10;). Then we proceeded to delete the resident *USO1* gene in the wild-type and mutant GHD overexpressing strains. As expected, neither of the resulting pair of strains was able to grow on a medium with glucose as the only carbon source (*Figure 9E*, lanes 2 and 6). Notably, the strain expressing E6K/G540S GHD as the sole Uso1 source grew essentially as the wild-type at 30 °C and, although debilitated, was viable at 37 °C, showing a substantial improvement of the growth capacity displayed by the gene-replaced mutant (*Figure 9*, lanes 7, 8, and 9). Thus, if expressed at sufficiently high levels, E6K/G540S GHD maintains viability at the optimal growth temperature.

Unexpectedly, the wild-type GHD also rescued the viability of the *uso1Δ* mutant when this was cultured with sucrose as a carbon source at 30°C and 37°C. In fact, at 30 °C, the *uso1Δ inuAp::GHD* strain grew like the wt (*Figure 9E*, lanes 3 and 4)(note that these experiments were carried out in a *RAB1+* background), suggesting that increased binding to a Golgi receptor facilitated by mass action compensated for the loss of the coiled-coil region and associated dimerization. The E6K/G540S GHD would have gained an affinity for this receptor, explaining why the doubly substituted GHD suppressed mis-localization of Uso1-GFP when RAB1 is compromised, even when its steady-state levels were very low. Forced expression, combined with a potentially increased binding affinity of E6K/G540S GHD to such a hypothetical receptor might be toxic.

## Uso1 is an associate of the early Golgi SNARE machinery, with the double substitution E6K/G540S increasing this association

What is the nature of this hypothetical receptor? To address this question, we screened for interactors of Uso1 among proteins acting at the same functional level (consumption of COPII vesicles by the early Golgi) using a modified version of the S-tag co-precipitation approach that we used to characterize TRAPP complexes (*Pinar et al., 2019*; *Figure 10A*). We constructed strains expressing wild-type or mutant Uso1, tagged endogenously with the S-tag and, as negative unrelated control, BapH an effector of RAB11 acting in late steps of the secretory pathway (*Pinar and Peñalva, 2017*). Then, derivatives of these three strains co-expressing each of the candidate Uso1 GHD targets, tagged with HA3 (also endogenously), were constructed. The resulting panel (*Figure 10B*) was screened for HA-tagged proteins co-precipitating more efficiently with the E6K/G540S version of Uso1 than with the wild-type, and satisfying the criterium of not co-purifying with BapH. To this end, cell-free extracts of these strains were incubated with S-agarose beads that were recovered by centrifugation. Proteins associated with the S-baits were revealed by anti-HA western blotting.

Not every protein specifically co-purified with Uso1 baits was demonstrated by the results obtained with the COG component COG2, which did not associate with any of the three baits (*Figure 10C, immunopecipitation (IP) number (#) 10* ). In contrast, β-COP was a promiscuous non-specific interactor pulled down by all three baits (*Figure 10C, IP #11*). Notably, the screen identified the Golgi syntaxin Sed5 within the specific Uso1 associates (*Figure 10C, IP #1*), an association reported previously by others for both p115 and fungal Uso1 (*Allan et al., 2000*; *Sapperstein et al., 1996*). That the

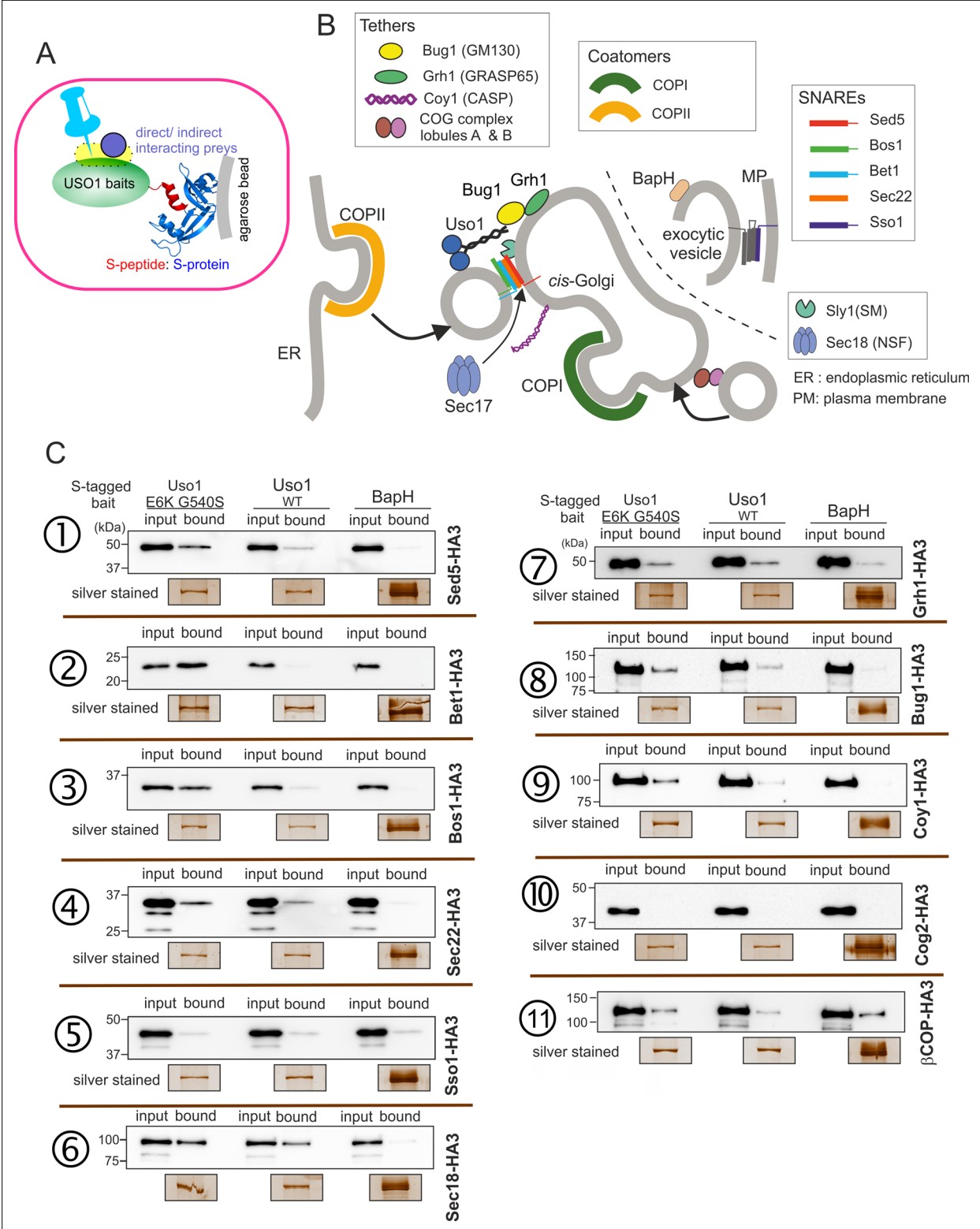

**Figure 10.** Screening the preferential association of proteins acting in the ER/Golgi interface with E6K/G540S Uso1. (**A**) S-tagged baits (Uso1, wt, and E6K/G540S, and the unrelated protein BapH), expressed after gene replacement, were captured with their associated polypeptides on S-protein agarose beads. Candidate associates, also expressed after gene replacement, were tagged with HA3. (**B**) Schematic depiction of the proteins listed in these experiments showing their sites of action. (**C**). Anti-HA3 western blot analysis of the indicated S-bait and HA3-prey combinations. Equal loading

*Figure 10 continued on next page*

*Figure 10 continued*

of Uso1 proteins was confirmed by silver staining of precipitates. Note that BapH, chosen as a negative control, is expressed at much higher levels than Uso1 proteins. Each panel is a representative experiment of three experimental replicates.

The online version of this article includes the following source data for figure 10:

**Source data 1.** Raw images for western blots and silver-stained gels and uncropped pictures with used exposures and regions indicated.

PM SNARE Sso1 did not interact at all with any of the S-baits demonstrated that Uso1 does not bind promiscuously to syntaxins (*Figure 10C, IP #5*). Importantly, Sed5 was brought down more efficiently by E6K/G540S Uso1, and not at all by BapH, even though levels of this unrelated bait, as assessed by silver staining of pull-downs, were markedly higher than those of either Uso1 version. Therefore, Sed5 (or its associates) might represent a potential anchor bound by E6K/G540S Uso1 with increased affinity to compensate for the lack of RAB1-mediated recruitment.

This analysis was extended to other members of the SNARE bundle forming in the ER/Golgi interface with the Qa Sed5: the Qb Bos1, the Qc Bet1, and the R-SNARE Sec22 (*Parlati et al., 2002*; *Pelham, 1999*; *Tsui et al., 2001*). Sec22 was slightly enriched in the E6K/G540S pull-down relative to wild-type Uso1 (*Figure 10C, IP #4*). The results with Bos1 and Bet1 were most noteworthy (*Figure 10C, IPs #2 and #3*) . Both were markedly increased in the E6K/G540S pull-downs, with Bet1 increasing the most. The AAA ATPase Sec18 disassembling *cis*-SNARE complexes also bound Uso1 and was slightly enriched in the sample of E6K/G540S associates, as was the Uso1-interacting Golgin Bug1, but not its membrane anchor Grh1 (*Figure 10C, IP #6-#8*) . We conclude that Uso1 is a component of the SNARE machinery and that this association is augmented by E6K/G540S.

S-tag-coprecipitations in *Figure 10* clearly singled out Bet1 (Qc) and Bos1(Qb) as the preys that were most strongly enriched with the mutant E6K/G540S bait relative to wild-type, and therefore with the highest probability of being direct interactors bound with greater affinity by the doubly substituted Uso1 mutant.

## Both G540S and E6K increase binding of the GHD domain to BOS1

Work by others implicated Uso1 in the assembly of the early Golgi SNARE bundle (*Sapperstein et al., 1996*). Thus, prompted by co-association experiments, we predicted that Uso1 would bind directly to Bet1, Bos1, and perhaps other SNAREs implicated in the biogenesis of the early Golgi. We anticipated that binding to Bet1 and Bos1 would be insufficient to recruit Uso1 to membranes in the absence of RAB1, but that once reinforced by E6K/G540S, Uso1 would not require RAB1 for its recruitment. To test this possibility, we searched for direct and E6K/G540S-enhanced interactions between the GHD and SNAREs with pull-down assays carried out with purified SNARE-GST fusion proteins as baits and Uso1-His6 constructs as preys. In all cases, the GST fusion proteins included the whole cytoplasmic regions of the SNAREs. *Figure 11—figure supplement 1* shows the AlphaFold2 predicted structure of the tetra-helical SNARE bundle.

Full-length Uso1 bound, weakly, to the Sec22 R-SNARE and to the Qb Bos1, and very efficiently [with *circa* 70% of the prey being pulled down (*Figure 11A and B*)] to the Qc Bet1. In contrast, Uso1 did not bind the Qa syntaxins tested, Sed5 and Sso1 (*Figure 11A*). The absence of interaction between Sed5 and Uso1, be it the wild-type or the E6K/G540S mutant version, strongly indicated that the association detected with S-tag pull-downs between Uso1 and Sed5 is bridged by other protein(s). This absence of binding cannot be attributed to Sed5-GST being incompetent for binding because Sed5-GST was competent in pulling-down highly efficiently its cognate SM protein Sly1, an interaction that did not occur with Sso1-GST (*Figure 11C*). Notably, the presence of the E6K/G540S double substitution in Uso1 (indicated with ** for simplicity in *Figure 11*) increased five times the amount of protein retained by the Qb Bos1 bait, whereas interaction with Bet1 did not change (*Figure 11A and B*). The double substitution in Uso1 did not promote interaction with Sed5 either. Thus, under normal circumstances, Uso1 is able to bind directly to three of the four SNAREs in the ER/Golgi interface, with binding to Bet1 being the strongest. The double E6K/G540S substitution increases binding to Bos1 very markedly and specifically, bringing it up to the levels of Bet1 without affecting, for example, binding to Bet1 or Sec22. Consistently, the GHD is sufficient to mediate interaction with Bet1 and Bos1, as well as, if E6K G540S-substituted, the increased binding of Bos1 to Uso1 (*Figure 11D and E*).

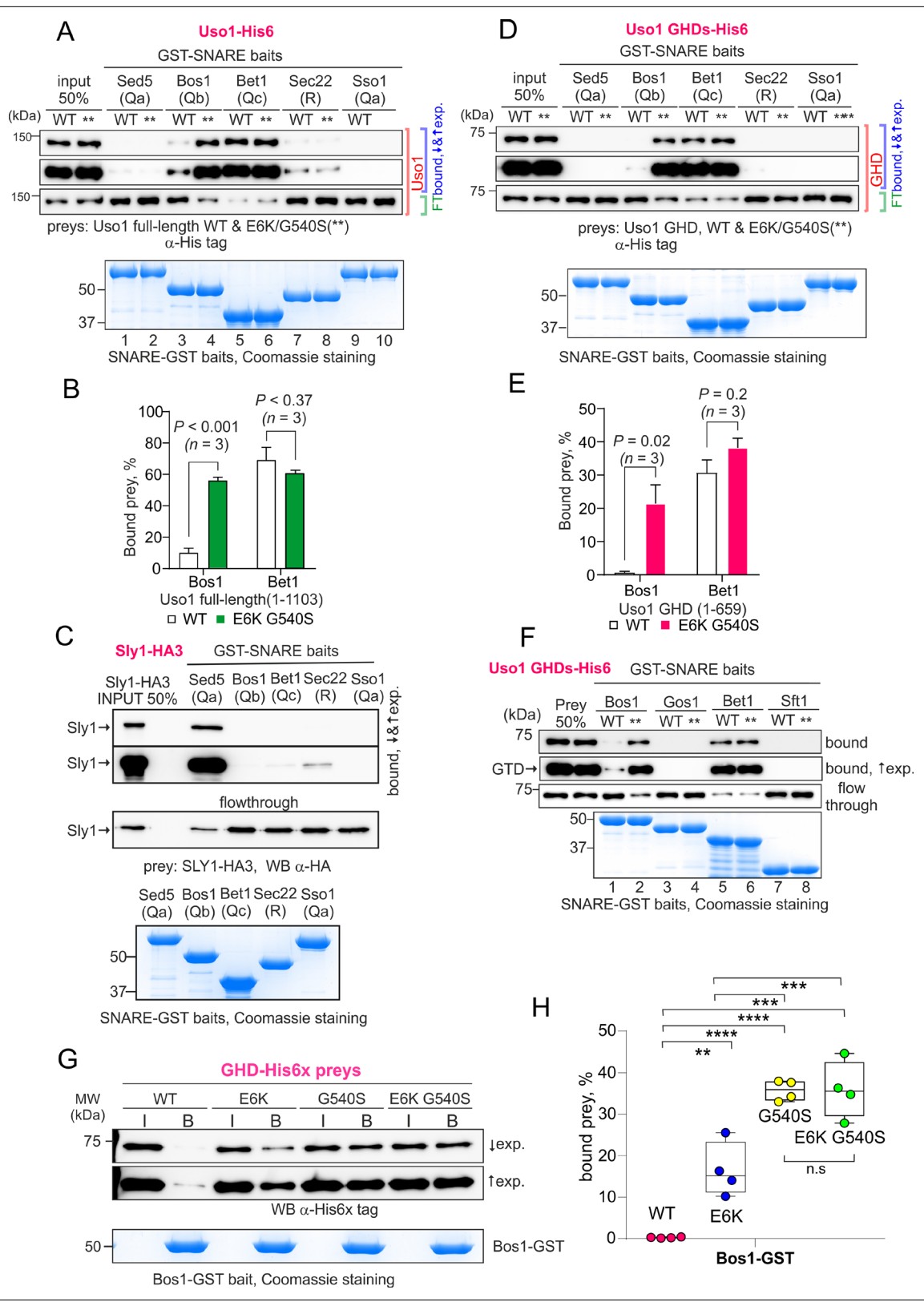

**Figure 11.** The Uso1 globular-head-domain (GHD) interacts directly with Bos1 and Bet1 SNAREs acting in the ER/Golgi interface. (**A**). Purified fusion proteins in which the complete cytosolic domains of the indicated SNAREs have been fused to GST were used in pulldown experiments with His-tagged, purified wild-type and E6K/G540S Uso1. The plasma membrane Qa syntaxin Sso1 was used as a negative control. Pulled-down material was analyzed by anti-His western blotting. In this and other panels ↑exp. and ↓exp. indicate high and low exposure to the chemiluminescent western blots,

*Figure 11 continued on next page*

*Figure 11 continued*

respectively. (**B**) Quantitation of the above experiment; significance was determined by unpaired *t*-student tests. Error bars represent SEM. (**C**) As in (**A**), but using in vitro synthesized, HA3-tagged Sly1 as prey. Samples were analyzed by anti-HA western blotting. (**D**) As in (**A**), but using wild-type and mutant GHD as preys, rather than full-length Uso1. (**E**). Quantitation of the experiment in (**D**). (**F**). GST pull-down experiment comparing the ability of the GHD to interact with the early Golgi Qb and Qc SNAREs (Bos1 and Bet1), with that of their medial Golgi counterparts (Qb Gos1 and Qc Sft1). (**G**) Bos1-GST pull-down experiments comparing single E6K and G540S mutants with the double mutant. (**I and B**) represent input and bound fractions. (**H**) Percentage of the input prey that is retained by the Bos1-GST beads. Data were analyzed by one-way ANOVA with Tukey's multiple comparisons. Whiskers indicate minimal and maximal values.

The online version of this article includes the following source data and figure supplement(s) for figure 11:

**Source data 1.** Raw images for western blots and Coomassie-stained gels and uncropped pictures with used exposures and regions indicated.

**Figure supplement 1.** AlphaFold2 prediction of the ER/Golgi SNARE bundle.

The GHD did not interact with Sec22, suggesting either that this R-SNARE is recruited by other parts of the protein or that binding is dimerization dependent.

Besides the Sed5/Bos1/Bet1/Sec22 combination, across cisternal maturation, Sed5 forms SNARE bundles in Golgi compartments located downstream of Uso1 domains (*Pelham, 1999*). In fungi, membrane fusion in the medial Golgi involves the Sed5 partners Gos1 (Qb) and Sft1 (Qc) substituting for Bos1 and Bet1, respectively, but neither Gos1 nor Sft1 bound wild-type or E6K/G540 Uso1 GHD (*Figure 11F*), demonstrating that interaction of Uso1 with Bos1 and Bet1 is highly specific. Therefore, it seems fair to conclude that increased binding for a SNARE receptor underlies the mechanism by which mutant Uso1 bypasses the need for RAB1 in the ER/Golgi interface.

Lastly, we tested the contribution of each substitution individually (*Figure 11, G and H*). Both E6K and G540S increased GHD binding to Bos1 markedly, relative to wild-type, but E6K was threefold less efficient in our pull-down assay, consisting with the respective ability to rescue *rab1* deficiency, and thus reinforcing the idea that increased Uso1-SNARE binding underlies the mechanism of suppression. However, we did not detect additivity in our pull-down assay, which is possibly attributable to the fact that Bos1-GST beads already pulls-down a substantial amount of the single G540S mutant. In any case, these data establish that the mechanism of suppression by E6K is unlikely to involve a role of the Glu6Lys-containing amphipathic alpha-helix in binding membranes.

In summary, Uso1 is an essential protein acting in the ER/Golgi interface, and we report here several important findings. We show that (i) the Golgi GTPase RAB1, which is essential for viability, becomes dispensable if there is an alternative method to recruit Uso1 to Golgi membranes; (ii) the coiled-coil region of Uso1 is dispensable to sustain viability, implying that the tethering role of the protein is not essential either, consistent with the non-essential roles of other Golgin tethers; (iii) the Uso1 GHD is essential for viability; (iv) the Uso1 GHD monomer, if present at suitably high levels, is sufficient to maintain viability; (v) that the Uso1 GHD is a direct and specific binder of the Qb SNARE Bos1, and a strong binder of the Qc Bet1; (vi) the mutations bypassing the need for RAB1 increase the affinity of the GHD for Bos1, indicating that *rab1Δ* viability rescue by E6K/G540S Uso1 occurs because SNARE anchoring provides an alternative mode of recruitment to Golgi membranes to that provided physiologically by RAB1.

## Interpretation of the mutations guided by AlphaFold2 predictions

To gain further insight into the mechanisms by which E6K and G540S increase the affinity of the Uso1 GHD for Bos1 we used AlphaFold2 to model the Uso1$^{GHD}$ domain alone, or together with each of the individual SNAREs, with both Bos1 and Bet1 simultaneously, and with RAB1.

Consistent with biochemical data, in the predicted model Bet1 and Bos1 interact with a medial and a C-terminal region, respectively, of Uso1$^{GHD}$. (*Figure 12*, *Figure 12—figure supplements 1 and 2*). Bet1 docks against a region of the GHD that is not affected by the mutations. In contrast, Gly540 and its environs dock against the N-terminal, triple α-helical Habc domain of Bos1. AlphaFold2 predicts that Uso1$^{GHD}$ interacts with Bos1 through a surface composed of the N-terminal part of the first Bos1 Habc α-helix and the loop between α-helices 2 and 3 (*Figure 12A and B*). The binding surface in Uso1$^{GHD}$ predictably involves the second α-helices of the ARM10 and ARM11 repeats, (α-helices 26 and 29), and the loop connecting the first two α-helices of ARM10 (*Figure 12B*). In AlphaFold2 models, Uso1 Gly540 (wild-type) is located at the beginning of Uso1$^{GHD}$α-helix 29, at the heart of the interaction surface, contributing to the Uso1-Bos1 interaction by coordinating the amide group of the Uso1

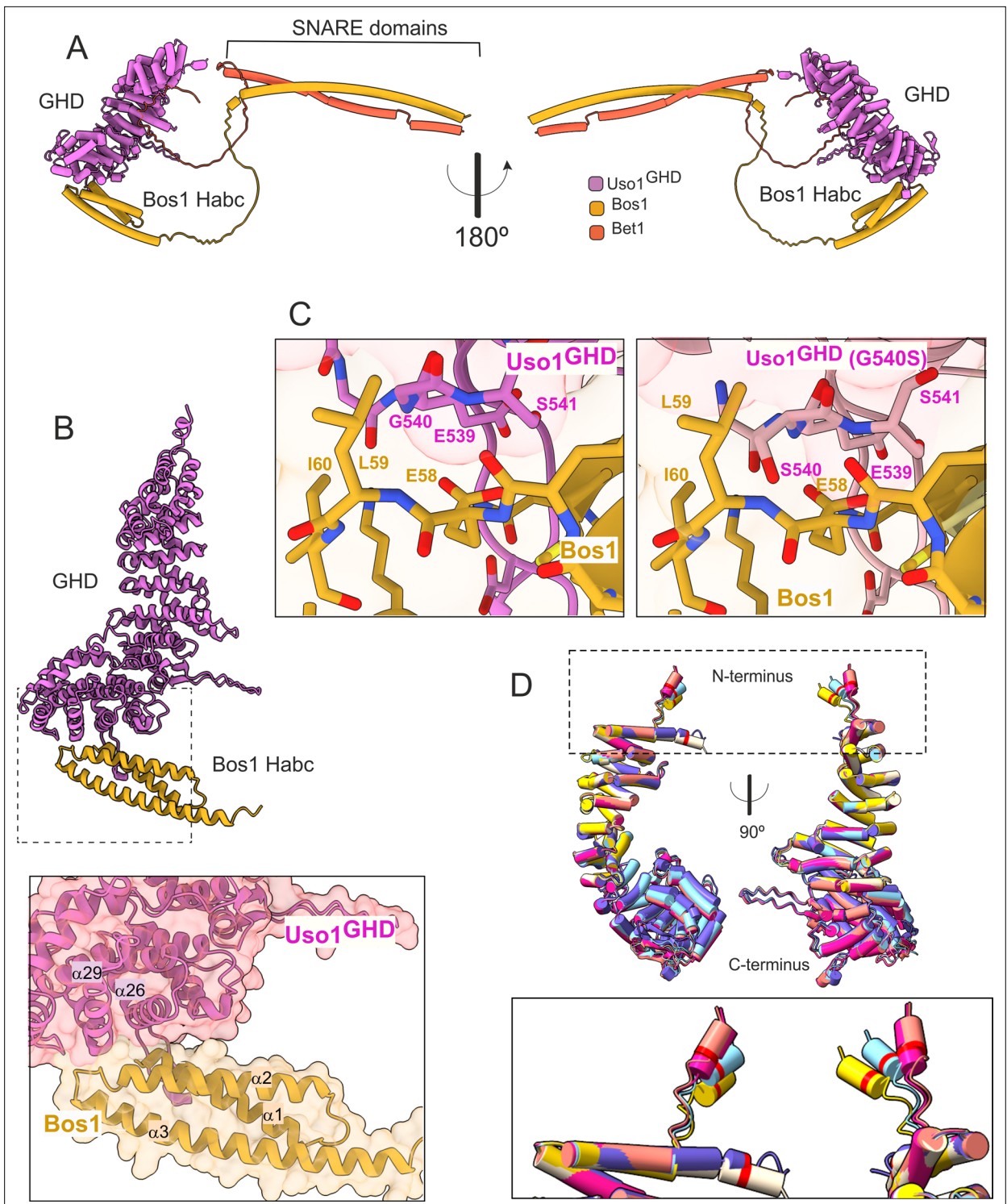

**Figure 12.** AlphaFold2 models provide insight into the mechanisms of suppression by E6K and G540S. (**A**). Model of full-length Uso1 bound to the ER/Golgi SNAREs Bos1 and Bet1. (**B**). Top, ribbon representation of the Bos1 N-terminal Habc domain and Uso1 globular-head-domain (GHD). Bottom, Inset combining surface and ribbon depiction. (**C**). Increased binding of Bos1 to G540S Uso1 appears to involve the insertion of Ser540 into a pocket located in the Habc domain of the Qb SNARE. Partial view of the Bos1-Uso1 GHD surface of interaction in the wild-type (left) and mutant (right) models. G540 and S540 are annotated. (**D**). The N-terminal amphipathic α-helix of Uso1 comprising the E6K substitution lies within a flexible stretch of the protein that might facilitate its insertion into membranes. Alignment of six independent predictions, each depicted in a different color, with Glu6 highlighted as a red collar surrounding the cylinders of the N-terminal α-helix. The Uso1 GHD was modeled alone, in a complex with SNARE proteins or with Ypt1. The N-terminal α-helix (boxed) adopts different positions, suggesting high flexibility.

*Figure 12 continued on next page*

*Figure 12 continued*

The online version of this article includes the following figure supplement(s) for figure 12:

**Figure supplement 1.** AlphaFold2 prediction of the Bet1-GHD interaction.

**Figure supplement 2.** AlphaFold 2 predictions of the RAB1 binding site on the Bet1/Bos1/Uso1 globular-head-domain (GHD) complex.

**Figure supplement 3.** Quality control assessment of AlphaFold2 predictions for the indicated complexes.

backbone with the Bos1 Glu58 carboxylate to create a hydrogen bond (*Figure 12C*). According to the most confident prediction, Gly540Ser results in the hydroxymethyl side chain protruding into a small pocket rimmed by the side chains of Bos1 Leu59 and Ile60, such that the Uso1 Ser540 hydroxyl group hydrogen-bonds the amide group of Bos1 Glu58 (*Figure 12C*). Besides creating a new hydrogen bond, PISA calculations indicate that Gly540Ser increases the surface of interaction ~10 Å$^2$, from ~824 Å$^2$ to ~831 Å$^2$. While this increase is not translated in a commensurate decrease in the solvation-free energy ($\Delta^i$G), which in fact it is slightly higher in the mutant (*Supplementary file 1*), this estimation does not consider the effect of satisfied hydrogen bonds and salt bridges across the mutant's interface. Predicted rearrangements induced by Gly540Ser involved the formation of three additional hydrogen bonds, besides that between Ser540 and Glu58. Assuming that each of the four-newly formed hydrogen bonds contributes approximately 0.5 kcal per mole to the free energy and that each of the four salt bridges in each model contributes approximately 0.3 kcal/mol, a total binding energy of –8.4 kcal/mol, and –9.6 kcal/mol was estimated for wild type and G540S Uso1 binding to Bos1 (*Supplementary file 1*).

In the case of the strong Uso1$^{GHD}$-Bet1 interaction (*Figure 12—figure supplement 1*), the N-terminal region of Bet1 consisting of *circa* 80 amino acids is disordered, and the pLDDT score of the different models is understandably low. However, all structural models depicting Bet1 interacting with Uso1$^{GHD}$ (e.g. in the context of the whole SNARE complex, Bet1 alone, or the isolated Bet1 N-terminal) consistently show a region where the pLDDT is higher. This region is predicted to form a kink in this N-terminal part of Bet1 that protrudes into the surface created by the α-helices 13 and 16 of Uso1 ARM4 and ARM5 (*Figure 12—figure supplement 1A*). Surface representations show that this section of the Bet1 polypeptide covers ~1300 Å$^2$ of the GHD, docking against the same side of the boomerang-shaped solenoid as Bos1 (*Figure 12—figure supplement 1A*), which is consistent with the orientation that these SNAREs should take during the formation of the SNARE pin (*Figure 12—figure supplement 2C*).

The mechanism by which Glu6Lys contributes to increasing the recruitment of Uso1 to the Golgi is unclear. This glutamate is located in a region with a low pLDDT score, which shows different conformations depending on the model (*Figure 12D*, boxed), indicating that it is flexible. Models concur in predicting the formation of a potentially amphipathic N-terminal α-helix containing Glu6, whose substitution by Lys (as in E6K) reinforces the positive charge of the polar face. However, the fact that E6K increases the binding of the GHD to Bos1 (*Figure 11G and H*) strongly argues against the possibility that E6K acts by increasing the recruitment of this amphipathic helix to membranes. Thus, an important question was whether this region represents a functional element that is required for the full function of Uso1. The N-terminal extension of *Aspergillus nidulans* Uso1 is 13 amino acids long. Its sequence is highly conserved among 196 Uso1 homologs of members of the class Eurotiomycetes, with several positions including GLU6 being invariant —In fact, sequence conservation across Eurotiomycetes in this region is even higher than in the first armadillo repeat (*Figure 13A*). This sequence similarity suggests that this extension plays a physiological role. Paradoxically, this conservation is lost in other classes of Ascomycota (including Saccharomycetes) and Basidiomycota (*Figure 13B and C*). However, circumstantial evidence supporting that this region contains a previously unnoticed and potentially functional secondary structure element is provided by AlphaFold, which predicts the presence of an N-terminal alpha-helix in equivalent position of Uso1 proteins from model organisms such as *Schizosaccharomyces pombe*, *Drosophila melanogaster*, and *Arabidopsis thaliana* (*Figure 13—figure supplement 1*), buttressing the idea that this structural element plays a physiological role.

To confirm this role, two sensitive genetic tests were used. They compared the ability of wild-type Uso1 with that of a mutant protein lacking the 13 N-terminal residues [Uso1(Δ1–13)] to rescue the lethality resulting from *uso1Δ* and *rab1Δ* mutations, respectively, under different expression levels, exploiting the regulation of the *inuAp* promoter by different carbon sources. As determined in control

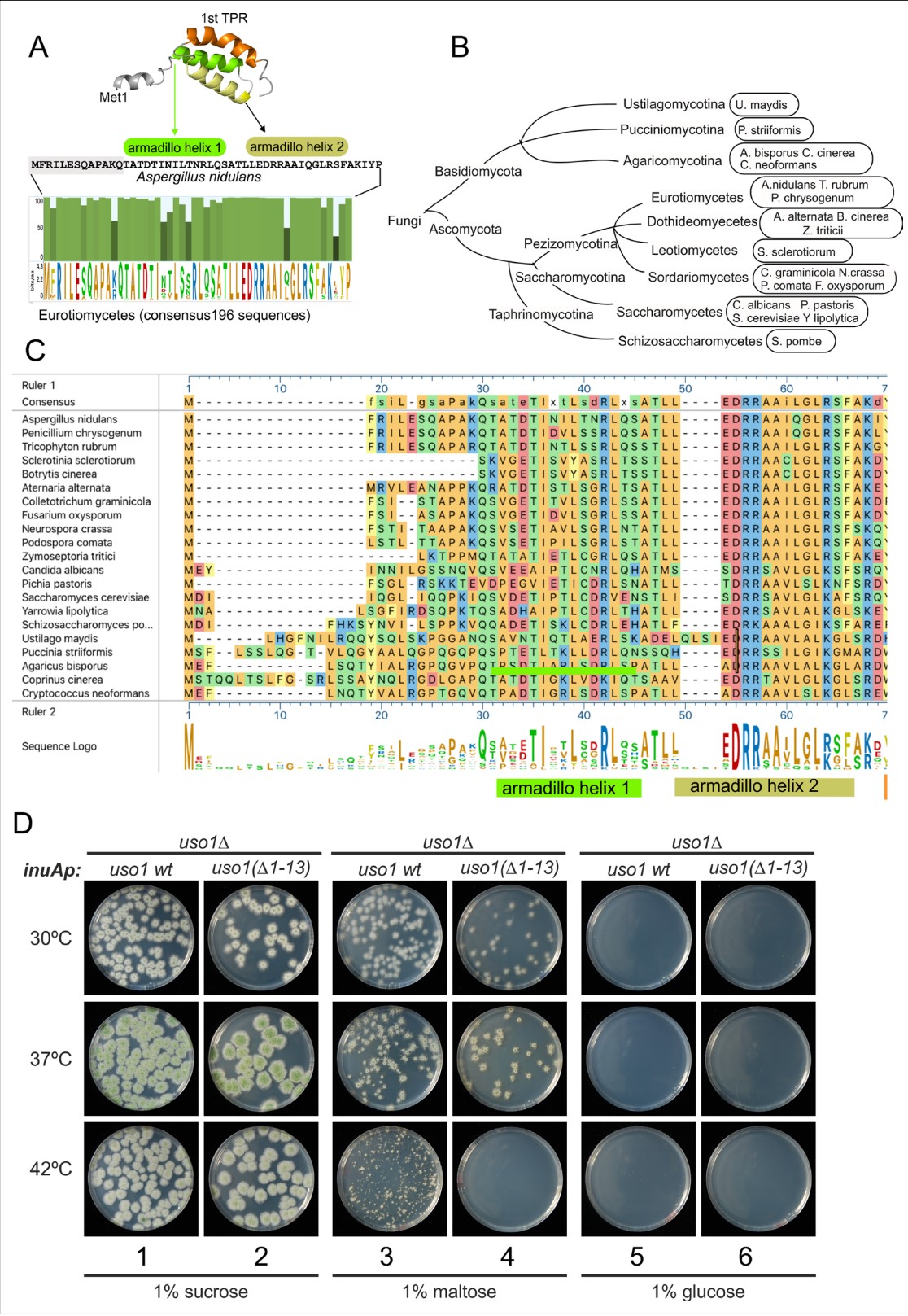

**Figure 13.** The N-terminal region of *Aspergillus* Uso1 containing Glu6 is required for full function. (**A**) Sequence comparison of N-terminal regions up to the second helix of the first armadillo repeat including 196 Uso1 proteins of Eurotiomycetes available in https://fungidb.org/fungidb/app (**B**) Phylogenetic tree of the Fungi with the indication of species that were used for the alignment in panel (**C**). (**C**) Multiple sequence alignment of

*Figure 13 continued on next page*

*Figure 13 continued*

indicated Uso1 sequences, constructed with ClustalW and Mega Align Pro of the DNAstar package. (**D**) The 13 N-terminal amino acids of Uso1 are required for full function (see text and *Figure 13—figure supplement 2*).

The online version of this article includes the following source data and figure supplement(s) for figure 13:

**Figure supplement 1.** AlphaFold prediction of N-terminal helix: model organisms.

**Figure supplement 2.**

**Figure supplement 2—source data 1.** Raw images for western blots and uncropped pictures with used exposures and regions indicated.

experiments by α-InuA-HA3 Western blotting (*Figure 13—figure supplement 2A*), the full expression of the promoter on sucrose results in a 23-fold higher levels of InuA than on glucose, but the expression on maltose increased InuA levels only threefold relative to glucose. Equipped with wild-type and (Δ1–13) single-copy transgenes integrated at the *inuA* locus (*Figure 13—figure supplement 2B*), we first demonstrated that the (Δ1–13) N-terminal deletion impaired complementation of a deletion allele of *uso1* (*Figure 13D*). In these experiments, pyrimidine-requiring *pyrG−* strains carrying *inuA-p::Δ(1-13) Uso1* or *inuAp::Uso1* transgenes were deleted for either *rab1* or *uso1*, substituting their coding regions by the *pyrG+* gene of *A. fumigatus* (*Figure 13—figure supplement 2B*). Even though *uso1Δ* and *rab1Δ* are lethal, the transformation of multinucleated protoplasts results in the formation of viable heterokaryotic colonies, in which the deletion alleles are in heterosis with the corresponding wild-type allele. Heterosis is resolved by plating conidiospores, in which only one nucleus segregates, onto media carrying or not pyrimidines (*Figure 13—figure supplement 2C*). Whereas untransformed nuclei were able to support growth only in the presence of pyrimidines, conidiospores of the transformed *uso1Δ* nuclei (prototrophs for pyrimidines) were rescued by both the wild-type and the N-terminally deleted Uso1 proteins at the three tested temperatures (30 °C, 37°C, and 42°C) when cultured on sucrose (full expression) (*Figure 13D*, lanes 1–2). In contrast, they were not rescued at all when cultured on glucose (low, basal expression, *Figure 13D*, lanes 5–6). We reasoned that high-level expression on sucrose could compensate for a weak loss-of-function resulting from *uso1 (Δ1–13)*. When cultured on maltose the wild-type complemented *uso1Δ* much less efficiently than under full expression conditions, but permitted growth even at 42 °C. In sharp contrast, Uso1(Δ1–13) was unable to rescue lethality at 42 °C (*Figure 13D*, lanes 3–4). Because western blotting indicated that the wild-type and the N-terminally deleted proteins were similarly expressed (*Figure 14A*), these data represent evidence that the N-terminal residues of Uso1 are required for the full function of the protein, playing a positive role.

To confirm these conclusions further, we tested rescue of the lethality resulting from *rab1Δ* by Uso1, Uso1(Δ1–13), and Uso1(E6K), using *inuAp* and heterokaryon rescue as described above. As expected, none of the three proteins was able to rescue *rab1Δ* on glucose at any tested temperature (*Figure 14B–D*, minus pyrimidine rows). In contrast, the wild-type was able to rescue *rab1Δ* lethality at 26 °C when cultured on sucrose (*Figure 14B*, lane 5), a partial suppression that was not achieved by Uso1(Δ1–13) (*Figure 14C*, lane 5), indicating that the N-terminal deletion impairs Uso1 function. Notably, Uso1(E6K) was an even stronger suppressor than the wild-type, permitting growth at 26°C and 30°C (*Figure 14D*, lane 5 and 6). Thus, according to these tests, the N-terminal deletion allele is a weak hypomorph, whereas the E6K-substituted allele is a weak hypermorph.

We additionally modeled RAB1 binding to the GHD in the absence and presence of Uso1 binders Bos1 and Bet1. AlphaFold2 predicts that the GHD interacts with RAB1 through a binding surface formed by the ARM3-5. repeats. On the other hand, the interactive region of RAB1 conforms to canons, as Uso1 α-helices 8 and 9 (ARM3) is predicted to interact with the Switch II region while α-helices 11 (ARM-4) and 14 (ARM-5) interact with the (Switch I) (*Figure 12—figure supplement 2A–C*). Importantly, the model indicates that within the Uso1-GHD jai alai basket, RAB1 binds to the opposite (convex) side of the Bet1 interacting area at the concave side, and away from the C-terminal helix where the Bos1 Habc domain predictably binds, which would allow Uso1 to bind these three interactors simultaneously (*Figure 12—figure supplement 2C*) (AlphaFold structures colored by confidence are shown in *Figure 12—figure supplement 3*). In addition, the predicted models supported a highly suggestive but as yet speculative implication: Uso1-RAB1 would be in an orientation that facilitates the docking of RAB1-loaded ER-derived vesicles with an acceptor membrane where SNARE zippering occurs (see discussion).

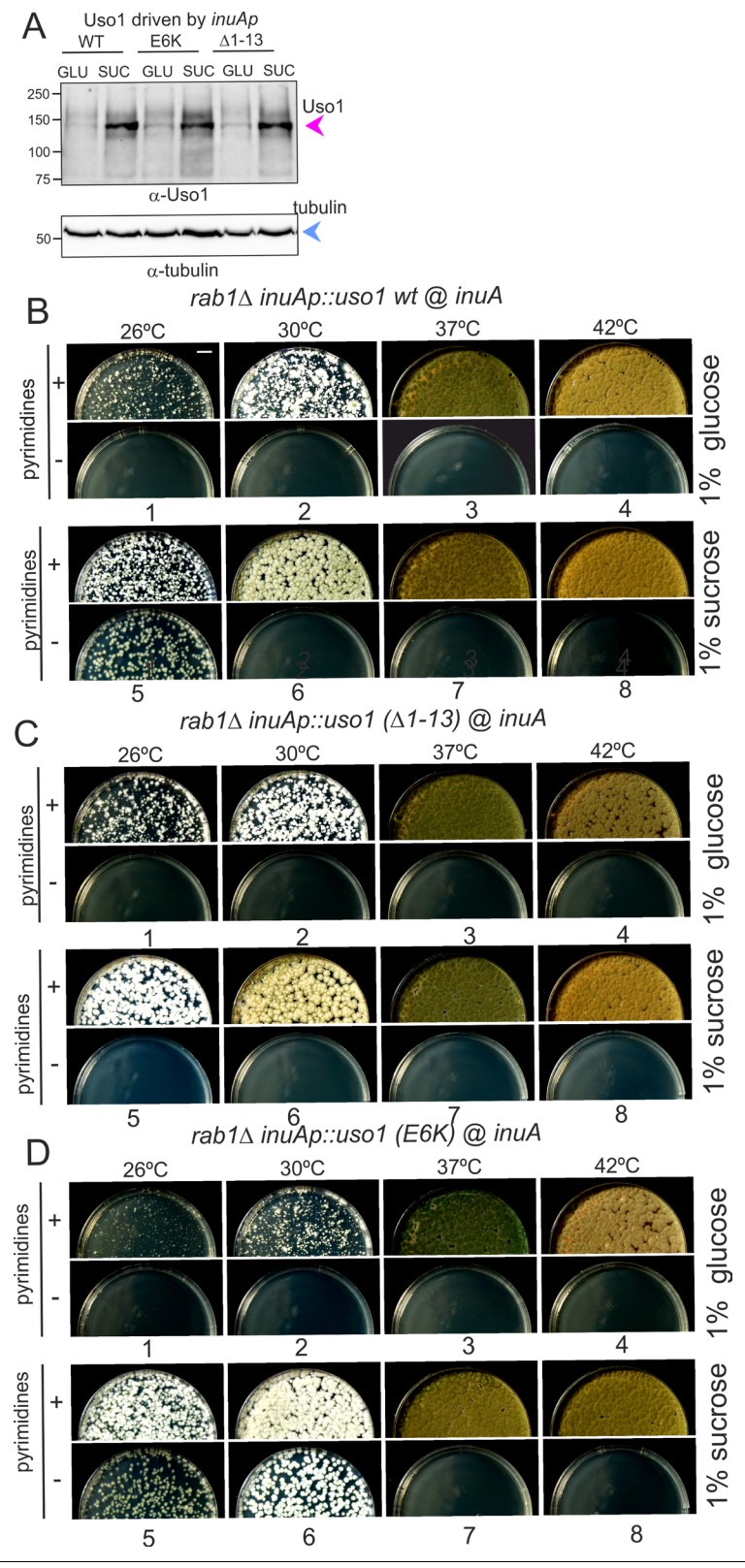

**Figure 14.** Over-expression of Uso1, but not of Uso1 (Δ1–13) rescues the lethality of *rab1Δ*, and rescue is stimulated by E6K. (**A**) Western blot showing that all three alleles used in growth tests shown below result in similar levels of protein when driven by the *inuAp* on sucrose (suc). The protein detected on glucose (glu) corresponds to the endogenous gene. (**B**) Overexpression of Uso1 wild-type permits growth of *rab1Δ* at 28 °C (**C**) Overexpression

*Figure 14 continued on next page*

*Figure 14 continued*

of Uso1 (Δ1–13) does not rescue *rab1Δ* at any temperature,. (**D**) Overexpression of Uso1 E6K permits growth at 26°C and 30°C.

The online version of this article includes the following source data for figure 14:

**Source data 1.** Raw images for western blots and uncropped pictures with used exposures and regions indicated.

## Discussion

RAB1 regulates transport at the ER/Golgi interface. Using an unbiased forward genetic screen to identify subordinated genes accounting for its essential role, we isolated two extragenic mutations resulting in single-residue substitutions in the RAB1 effector Uso1, usually regarded as a tether. The single-residue substitutions lie at opposite ends of the jai alai basket-shaped GHD; both are individually able to rescue the viability of *rab1Δ* strains at 30 °C and, when combined, even at 42 °C.

Subcellular localization experiments hinted at the mechanism by which the double mutation rescues *rab1Δ* lethality. Uso1 plays its physiological role in an early Golgi compartment, characterized by containing the Golgi-ER receptor Rer1. Uso1 delocalizes to the cytosol in a RAB1-deficient background. Under normal circumstances, the Uso1 CTR acts in concert with RAB1 to recruit the protein to the Golgi. Genetic evidence (*Figure 8*) showed that this contribution requires the Golgi-localized tether composed of the membrane anchor Grh1 and its associated golgin BUG1 (*Behnia et al., 2007*), homologs of human GRASP65 and GM130, respectively. In the absence of RAB1, engagement of the CTR with BUG1 is insufficient to stabilize wild-type Uso1 on membranes. However, the double E6K/G540S restores Uso1 to Golgi structures, suggesting that mutant Uso1 GHD had gained an affinity for another element.

Inspired by cross-linking studies that identified Sed5, membrin (Bos1) and mBet1 as weak p115 interactors (*Allan et al., 2000*), directed S-tag co-precipitation analysis of cell extracts showed that E6K/G540S increases association of Uso1 with four SNAREs, Sed5, Bos1, Bet1, and Sec22, mediating fusion events at the ER/Golgi interface, and with the SNARE regulator Sec18, indicating that Uso1 is a component of the SNARE fusion machinery. Pull-down assays using purified proteins showed that wild-type Uso1 binds directly to Bet1, Bos1 (weakly), and Sec22 (*Figure 11*). The RAB1 suppressor mutations markedly increase the binding of Uso1 to Bos1, and we note that the magnitude of this increase correlates strictly with the degree of *rab1Δ* suppression. For example, E6K binds better than wild-type to Bos1, and G540S binds even better (*Figure 11*). The latter is a better suppressor than the former; Uso1 overexpression suppresses weakly *rab1Δ*, whereas an analogous construct carrying E6K suppresses more strongly (*Figure 14*). While the most parsimonious interpretation of our data is that Uso1 mutants suppress the loss of RAB1 by binding more tightly to SNAREs, we acknowledge that this conclusion is not formally established, as we cannot rule out that the mutations result in additional changes in the Uso1/RAB1/SNARE module that represent the primary cause for suppression.

The Uso1/p115 family has been previously implicated in the regulation of SNARE complexes by 'catalyzing' the formation of the SNARE bundle (*Allan et al., 2000*; *Shorter et al., 2002*; *Wang et al., 2014*). These models rely on the identification of a region in the CCD that resembles a SNARE domain and that is capable of pairing with actual SNAREs to form intermediates that would precede the formation of the tetra-helical complex. These models did not reveal a role of the GHD, and indeed (*Shorter et al., 2002*) discuss that 'intriguingly, the GHD of p115 seems to play no direct role... in the assembly of the SNAREpin.' Our results add another layer of complexity by showing that the GHD plays a positive role in an as yet undefined manner in the machinery that mediates the fusion of ER-derived vesicles with the Golgi, such that increased binding of the GHD to SNAREs compensates for even the complete loss of RAB1 (*Figures 1 and 14*). We notice, however, that our conclusions do not discard the possibility that the GHD and the CCD cooperate in the assembly of the SNARE bundle.

The ability of Uso1 to mediate long-distance tethering of vesicles was initially considered the essential role of this protein. In *S. cerevisiae*, *uso1-1* truncates the protein after residue 950, leaving approximately one-third of the CCD. These 950 residues are sufficient to retain some function (*Seog et al., 1994*). Truncation further upstream results in a complete loss-of-function, which was taken as evidence that the coiled-coil domain is essential (*Seog et al., 1994*). However, this work did not address whether severe truncation of the protein affected stability. Our data show that levels of a protein deleted for the CCD are markedly reduced, and that elevating levels by forced expression, or

introducing E6K/G540S in the truncated gene by allele replacement substantially rescues the Uso1 function. These data support our contention that the essential function of Uso1 involves SNARE-mediated fusion, rather than long-distance tethering.

Genetic interactions between Uso1/p115, RAB1, and SNAREs have been studied in *S. cerevisiae* (*Dascher et al., 1991*; *Sapperstein et al., 1996*). These studies were based on rescuing lethality resulting from a deficit of Uso1 or RAB1/Ypt1. Suppressors can be classified into two classes. Class I are alleles overexpressing SNAREs, which suppress *Uso1* and *RAB1* hypomorphic alleles and, weakly, *uso1Δ* and *rab1Δ*; notably the two SNAREs suppressing *uso1Δ* are Sec22 and Bet1, i.e., those bound by *Aspergillus w*ild-type Uso1; Class II, involving a mutant protein expressed at normal levels, had only one representative, *SLY1-20*, a modification-of-function allele in the gene encoding the ER/Golgi SM protein that rescues viability of *rab1Δ*. *SLY1-20* promotes the open conformation of Sed5, thereby enhancing the formation of the SNARE complex (*Demircioglu et al., 2014*). *Aspergillus* E6K/G540S Uso1 is the second member of this class, substantially rescuing *rab1Δ* at physiological levels of expression (*Figure 1D*).

What regions of Uso1 are required for its function? We exploited the modification-of-function associated with E6K/G540S to address this question: Analytical ultracentrifugation studies on the oligomerization status of the different domains of the protein expressed and purified from bacteria strongly indicated that the ability of Uso1 to dimerize resides in the CCD, and that the GHD is monomeric. While results with bacterially-expressed proteins should be interpreted cautiously, we note that the GHD in extracts of *Aspergillus* hyphae also behaves as a monomer in gel filtration experiments (*Figure 9*). Our data show that a GHD polypeptide carrying E6K/G540S is sufficient to support viability when expressed as the only source of Uso1, and that it appears to do so as a monomer. These conclusions shift the central role in the Uso1 function attributed to the CCD towards the globular domain, or in mechanistic terms from tethering to SNARE regulation.

AlphaFold2 modeling of the regions of the interaction of the GHD with Bos1 and Bet1 predicted that, of the two Uso1 residue substitutions, only Gly540Ser maps to the region mediating the interaction with Bos1 whereas, as indicated by GST pull-downs, neither affected the Bet1-interacting region, which is predicted to involve a large surface, consistent with its strong 'constitutive' binding and its likely role as the major physiological receptor for Uso1. In the case of substitutions affecting Bos1 binding, Gly540 substitution by Ser in the GHD results in the formation of four additional hydrogen bonds, of which one involves the side-chain of Ser540 and the remaining three result from rearrangements in the pocket in which this residue is inserted (*Supplementary file 1*). As for the Glu6Lys substitution, pull-down experiments clearly show that it increases the avidity of the GHD for Bos1, yet it lies at the opposite end of the GHD where the SNARE is predicted to bind. AlphaFold models Glu6 in the charged surface of an amphipathic alpha helix located in the 13-residue N-terminal extension that precedes the first armadillo repeat. Sequence conservation within this extension is restricted to the closest fungal relatives. However, genetic evidence strongly indicates that it is required for the full function of Uso1, and AlphaFold predicts that this region contains a small alpha-helix in organisms as distant as *Caenorhabditis elegans*, *Arabidopsis thaliana,* and *Schyzosaccharomyces pombe*. How this substitution affects Uso1-Bos1 binding at a distance will require further investigation. At present, the interpretation, based on minor changes in the spatial arrangement of armadillo helices suggested by the predictor (*Figure 12D*), that Glu6Lys might affect Bos1 binding at a distance remains highly speculative.

In summary, our work establishes that the essential role of Uso1 resides not in its CCD domain tethering donor and acceptor membranes, but in the GHD. When expressed at sufficient levels, or when its activity is boosted by E6K/G540S, the GHD is capable of substantially complementing *uso1Δ*, which we take as definitive evidence that the tethering function of Uso1 is not required for the protein to play its essential function. Therefore, our conclusions add another layer of evidence supporting the view that Uso1 critically cooperates with SNAREs.

## Ideas and speculation

While the molecular details of this mechanism will be addressed in the future, it is tempting to speculate that the GHD contributes, with Sly1, to orientate SNAREs to form a productive bundle, acting as chaperones, similarly to the HOPS SM component Vps33, which appears to align SNAREs in a pre-zippering stage, facilitating their assembly (*Baker and Hughson, 2016*; *Baker et al., 2015*; *Ren et al.,*

*2009*; *Yu and Hughson, 2010*; *Zhang and Hughson, 2021*; *Zhang and Yang, 2020*). Sec39, a subunit of the Dsl1 tethering complex, like the GHD an α-solenoid, binds the Qc-SNARE Ufe1, whereas Tip20, the other 'leg' of Dsl1, binds the Habc domain of the Qb SNARE Sec20. The authors hypothesize that Dsl1 arranges the Qb and Qc in an open conformation, presenting them to Sly1 already associated with syntaxin Ufe1 and R-SNARE Sec22, thus acting as a chaperone of the whole trans-SNARE complex formation (*Travis et al., 2020*). The Uso1 GHD could behave in a similar manner to Dsl1 facilitating the interplay between paired Bos1-Bet1 with the Sed5-Sec22-Sly1 cluster.

A second, also speculative interpretation is that tethering occurs in two steps, with golgins acting at long distances, approximating vesicles to the vicinity of SNAREs. Then SNAREs would engage RAB1–Uso1 to serve as short-range tether preceding the zippering up of membranes. The >700 Å-long Uso1 CCD would cooperate with Bug1 in the first step, whereas the Uso1 GHD would cooperate with RAB1 and Bet1/Bos1/Sec22 in the second, exploiting the fact that GHD binders use surfaces located at opposite sides of the α-solenoid (*Figure 12—figure supplement 2*). Such arrangement implies that RAB1 C-terminal isoprenoids inserted into the donor vesicle membrane would be *circa* 220 Å apart from the C-termini of the SNAREs inserted in the acceptor membrane. Two-step tethering might impose one additional level of specificity, preventing unproductive fusion events mediated by other SNARES circulating through the ER/interface and directing, by way of Uso1 interactions, incoming vesicles to fusion-competent areas enriched in target SNAREs (*Bentley et al., 2006*).

## Materials and methods

### *Aspergillus* techniques

Standard *A. nidulans* media were used for growth tests, strain maintenance, and conidiospore harvesting (*Cove, 1966*). GFP−, HA3−, and S-tagged alleles were introduced by homologous recombination-mediated gene replacement, using transformation (*Tilburn et al., 1983*) of recipient *nkuAΔ* strains deficient in the non-homologous end joining pathway (*Nayak et al., 2006*). Complete strain genotypes are listed in *Supplementary file 2*. These alleles were usually mobilized into the different genetic backgrounds by meiotic recombination (*Todd et al., 2007*).

Null mutant strains were constructed by transformation-mediated gene replacement, using as donor DNA cassettes made by fusion PCR (primers detailed in *Supplementary file 3*) carrying appropriate selectable markers (*Szewczyk et al., 2006*). Integration events were confirmed by PCR with external primers. When allele combinations were expected to be synthetically lethal or severely debilitating, the corresponding strains were constructed by sequential transformation, but the second such manipulation was always carried out using *pyrG*$^{Af}$ as a selective marker, which favors the formation of heterokaryons in which untransformed nuclei supported growth (*Osmani et al., 1988*). Conidiospores, in which single nuclei had segregated (i.e. homokaryotic nuclei), were scrapped and streaked onto plates carrying a doubly-selective medium. Absence of growth or appearance of microcolonies, combined with a positive PCR diagnostic of heterokaryosis of the primary transformants, was taken as an indication of lethality. Whenever possible, colony PCR of microcolonies was always used to genotype the desired genetic intervention.

The following proteins were C- or N-terminally tagged endogenously, using cassettes constructed by fusion PCR (*Nayak et al., 2006*; *Szewczyk et al., 2006*): Uso1-GFP and Uso1$^{E6K/G540S}$-GFP, Uso1-HA3 and Uso1$^{E6K/G540S}$-HA3; Uso1-S and Uso1$^{E6K/G540S}$-S; BapH-S (*Pinar and Peñalva, 2017*), Sec13-mCherry (*Bravo-Plaza et al., 2019*; *Bravo-Plaza et al., 2019*), Gea1-mCherry and Sec7-mCherry (*Arst et al., 2014*), mCherry-Sed5 (*Pantazopoulou and Peñalva, 2011*), mCherry-RAB1 (*Pinar et al., 2013*), HA3-Sed5, HA3-Bet1, HA3-Bos1, HA3-Sec22, Sec18-HA3, Grh1-HA3, Bug1-HA3, Coy1-HA3, COG2-HA3, and β-COP-HA3.

### Antibodies for western blotting

Antiserum against Uso1 was raised in rabbits by Davids Biotechnology. Animals were immunized with the Uso1 GHD (residues 1–659), and tagged with His6x. Recombinant expression in *E. coli* and Ni$^{2+}$ affinity purification are described below. Target antibodies were purified from raw antiserum by affinity chromatography through Hi-Trap NHS columns (#17-0716-01, Cytiva) charged with Uso1 antigen following the manufacturer's instructions. Affinity-bound antibodies were eluted with 100 mM glycine (pH 3.0), then neutralized with 2 M Tris to a pH of 7.5 and stored at −20 °C.

### *inuA* promoter-driven expression of Uso1 in *Aspergillus*

Different versions of Uso1 were expressed in a carbon source-inducible manner from the locus of the inulinase-encoding *inuA* (AN11778) gene. The gene replacement cassette was assembled through fusion PCR of four different elements (primers listed in *Supplementary file 3*) from 5' to 3': (1) *inuA* promoter, (2) cDNA sequence encoding Uso1 wt or mutant (E6K, G540S) GHD (residues 1–659), or full-length Uso1 and its mutant versions, (3) *Aspergillus fumigatus riboB* gene as selection marker, and (4) *inuA* gene 3'-flanking region.

A *pyrG*89, *nkuA*Δ::bar, *riboB*2 *Aspergillus nidulans* strain was transformed with this cassette, replacing the *inuA* gene. The promoter of *inuA* is inducible by sucrose and results in moderately high expression levels (*Hernández-González et al., 2018*). When using 1% (w/v) maltose as the sole carbon source, levels are three times higher than those with 1% (w/v) glucose, and eight times lower than those obtained with 1% (w/v) sucrose (*Figure 13—figure supplement 1A*). The resulting strains were subsequently transformed with the *uso1*Δ deletion cassette (*A. fumigatus pyrG* as selection marker) to eliminate endogenous expression of Uso1 wild-type protein. Similarly, those expressing full-length Uso1 or mutant versions were transformed with a *rab1*Δ deletion cassette, to test if they were able to rescue *rab1*Δ lethality.

## Plasmids for protein expression in *E. coli*

### His6x-tagging constructs

#### pET21b-Uso1-His6x and pET21b-Uso1(E6K/G540S)-His6x

cDNA encoding full length Uso1 (residues 1–1103) was cloned into a pET21b *Nde*I/*Not*I linearized vector.

#### pET21b-Uso1ΔCTR-His6x and pET21b-Uso1(E6K/G540S)ΔCTR-His6x

lacking the C-Terminal Region of the coiled-coil Domain (residues 1–1040) pET21b-Uso1GHD-His6x and pET21b-Uso1(E6K/G540S)GHD-His6x: cDNA encoding the GHD of Uso1 (residues 1–659) was cloned as a *Nde*I/*Xho*I insert into a pET21b *Nde*I/*Xho*I linearized vector.

#### pET21b-Uso1 CCD-His6x

cDNA encoding Uso1 coiled-coil Domain (residues 660–1103) was cloned as a *Nde*I/*Not*I insert into a pET21b *Nde*I/*Not*I linearized vector.

### TNT expression plasmids

#### pSP64-Sly1-HA

this plasmid carries cDNA encoding full length Sly1 (AN2518) C-terminally tagged with a HA3x epitope, cloned as an *Nsi*I/*Sac*I insert into *Pst*I/*Sac*I pSP64(PolyA) vector

### GST-tagging constructs

In all cases, the complete cytosolic domain of SNAREs was used.

#### pET21b-Sed5-GST

cDNA encoding Sed5/AN9526 cytoplasmic domain (residues 1–322) C-terminally tagged with GST, was cloned as a *Nde*I/*Sal*I insert into a pET21b *Nde*I/*Xho*I linearized vector.

#### pET21b-Bos1-GST

cDNA encoding Bos1/AN11900 cytoplasmic domain (residues 1–219) C-terminally tagged with GST, was cloned as a *Nde*I/*Sal*I insert into a pET21b *Nde*I/*Xho*I linearized vector.

#### pET21b-Bet1-GST

cDNA encoding Bet1/AN5127 cytoplasmic domain (residues 1–144) C-terminally tagged with GST, was cloned as a *Nde*I/*Sal*I insert into a pET21b *Nde*I/*Xho*I linearized vector.

### pET21b-Sec22-GST

cDNA encoding Sec22/ASPND00903 cytoplasmic domain (residues 1–198) C-terminally tagged with GST, was cloned as a *NheI/SalI* insert into a pET21b *NheI/XhoI* linearized vector.

### pET21b-Sso1-GST

cDNA encoding Sso1/AN3416 cytoplasmic domain (residues 1–271) C-terminally tagged with GST, was cloned as a *NdeI/SacI* insert into a pET21b *NdeI/SacI* linearized vector.

### pET21b-Gos1-GST

cDNA encoding Gos1/AN1229 cytoplasmic domain (residues 1–208) C-terminally tagged with GST, was cloned as a *NdeI/SalI* insert into a pET21b *NdeI/XhoI* linearized vector.

### pET21b-Sft1-GST

cDNA encoding Sft1/AN10508 cytoplasmic domain (residues 1–73) C-terminally tagged with GST, was cloned as a *NdeI/SalI* insert into a pET21b *NdeI/XhoI* linearized vector.

## Co-precipitation experiments with total cell extracts

Preparation of *Aspergillus* total cell extracts was done as described, with minor modifications (*Pinar et al., 2019*; *Pinar et al., 2019*). 70 mg of lyophilized mycelium were ground with a ceramic bead in a Fast Prep (settings: 20 s, power 4). The resulting fine powder was resuspended in 1.5 ml of extraction buffer [25 mM HEPES-KOH (pH 7.5), 200 mM KCl, 4 mM EDTA, 1% (v/v) IGEPAL CA-630 (NP-40 substitute, #I8896, Sigma), 1 mM DTT, 2 µM MG-132 proteasome inhibitor (#S2619, SelleckChem) and complete ULTRA EDTA-free inhibitor cocktail (#5892953001, Roche)]. Approximately 0.1 ml of 0.6 mm glass beads were added and thoroughly mixed. This suspension was homogenized with a 10 s pulse at the Fast Prep (power 6) followed by a 10 min incubation at 4 °C. This homogenization step was repeated two times before clarifying the extract by centrifugation at 4 °C and 15,000 × *g* in a microcentrifuge. Total protein concentration of the extracts was determined by Bradford. Bovine Serum-Albumin (BSA) was then added as a blocking agent to the cell extract (final concentration 1% (w/v)). Binding reactions were carried out in 0.8 ml Pierce centrifuge columns (#89869, ThermoFisher): 9 mg of protein were mixed with 20 µL of S-protein Agarose beads (#69704, Novagen), that had been previously washed in extraction buffer with 1% (w/v) BSA. This buffer was also added to complete the final reaction volume of 0.6 ml. The mix was incubated for 3 hr at 4 °C in a rotating wheel. Columns were then opened at the bottom and gently centrifuged to remove the supernatant and collect the protein-bound beads. These were resuspended in the extraction buffer without inhibitors and incubated in rotation for 10 min at 4 °C, followed by two more washing steps in the extraction buffer without detergent and inhibitors. To elute proteins bound to the beads, 30 µl of Laemmli loading buffer [62.5 mM Tris-HCl (pH 6.0), 6 M urea, 2% (w/v) SDS, and 5% (v/v) β-mercaptoethanol] were added and the columns incubated at 90 °C for 2 min. The columns were centrifuged to collect the eluate, of which a 40% of the final volume was resolved in an SDS-polyacrylamide gel and then transferred to nitrocellulose for α-HA (#3F10, Roche) western blotting.

## Purification of Uso1 constructs tagged with His6x

Full-length His6-tagged Uso1, Uso1ΔCTR, Uso1 GTD, and Uso1 CCD constructs, wild-type and mutant versions, were expressed in *E. coli* BL21(DE3) cells harboring pET21b-His6 derivatives and pRIL. Bacteria were cultured at 37 °C in an LB medium containing ampicillin and chloramphenicol until reaching an $OD_{600nm}$ of 0.6. Then, IPTG was added to a final concentration of 0.1 mM. Cultures were shifted to 15 °C and incubated for 20 hr. Bacterial cells were collected by centrifugation and pellets were stored at –80 °C. For purification, frozen pellets were thawed in ice and resuspended in ice-cold bacterial cell lysis buffer [20 mM sodium phosphate buffer, pH 7.4, 500 mM KCl, 30 mM imidazole, 5% (v/v) glycerol, 1 mM β-mercaptoethanol, 1 mM $MgCl_2$, 0.2 mg/ml lysozyme and 1 µg/ml of DNAse I and cOmplete protease inhibitor cocktail (#11873580001, Sigma)]. This cell suspension was mechanically lysed in a French press (1500 kg/cm²) and the resulting lysate was centrifuged at 10,000 × *g* and 4 °C for 20 min to remove the cell debris. The supernatant was then transferred to polycarbonate tubes and centrifuged at 100,000 × *g* and 4 °C for 1 hr in an XL-90 ultracentrifuge (Beckman Coulter). 50 ml of cleared lysate was incubated with 400 µL of Ni-Sepharose High-Performance beads

(#17526801, Cytiva) for 2 hr at 4 °C. After this step, His-tagged protein-bound beads were pelleted at low-speed centrifugation and washed three times in lysis buffer [20 mM sodium phosphate buffer, pH 7.4, 500 mM KCl, 5% (v/v) glycerol, 1 mM β-mercaptoethanol] with increasing concentrations of imidazole. Finally, Ni2+ bound His6 proteins were eluted with 0.5 M imidazole buffer. 5 ml of eluted protein were loaded onto a HiLoad 16/600 Superdex 200 column (Cytiva) and run at 1 ml/min flow rate on an AKTA HPLC system, using phosphate buffered saline PBS containing 5% (v/v) glycerol and 1 mM β-mercaptoethanol. Fractions containing protein were pooled, analyzed for purity by SDS-PAGE followed by Coomassie staining, and finally quantified on a UV-Vis spectrophotometer before being stored at –80 °C.

## Purification of SNARE constructs tagged with GST

cDNAs encoding the cytosolic domains of SNAREs fused to a C-terminal GST were cloned into pET21b. Bacterial cultures and protein expression conditions were as described above for Uso1-His6 constructs. Frozen pellets were thawed in ice and resuspended in chilled bacterial cell lysis buffer [25 mM Tris-HCl (pH 7.4), 300 mM KCl, 5 mM MgCl$_2$, 1 mM DTT, 0.5 mg/ml lysozyme, 1 µg/ml of DNAse I and cOmplete protease inhibitor cocktail (#11873580001, Sigma)]. This cell suspension was incubated for 30 min in ice before being mechanically lysed in a French press (1500 kg/cm$^2$). The lysate was incubated for a further 30 min on ice and centrifuged at 20,000 × *g* and 4 °C for 30 min. After adding 10 mM EDTA to the clarified supernatant to stop DNAse I activity, it was transferred to a 50 ml tube, mixed with 500 µL of glutathione Sepharose beads 4B (#17075601, Cytiva) and rotated for 2 hr at 4 °C. After incubation, SNARE-GST-bound beads were pelleted by gentle centrifugation and washed three times for 10 min at 4 °C in 25 mM Tris-HCl (pH 7.4), 500 mM KCl, 5 mM EDTA, 1 mM DTT and subsequently transferred to a 0.8 ml Pierce column. Beads were washed six times (10 min at RT) in 200 µL of elution buffer [50 mM Tris-HCl (pH 8.0), 200 mM KCl, 10 mM glutathione, and 1 mM DTT]. These fractions were collected and pooled (~1 ml), then buffer-exchanged to storage buffer (PBS, 5% (v/v) glycerol, and 0.1 mM DTT) in a PD MidiTrap G-25 column. Protein concentration and purity were assessed by spectrophotometry and SDS-PAGE followed by Coomassie staining. Protein stocks were kept frozen at –80 °C.

## SNARE-GST pull-downs with purified Uso1-His6 constructs (Uso1, Uso1 GHD)

Binding reactions were performed in 0.8 ml Pierce centrifuge columns. 75 µg of purified SNARE-GST were mixed with 15 µL of glutathione Sepharose 4B beads and storage buffer to a final volume of 0.3 ml. Columns were rotated at 4 °C for 2 hr before the supernatant was removed after low-speed centrifugation. Subsequently, His6 preys were added to a final concentration of 0.2 µM in a total volume of 0.4 ml of pull-down binding buffer [25 mM HEPES-KOH (pH 7.5), 150 mM NaCl, 10% (v/v) glycerol, 0.1% (v/v) Triton X-100, and 0.1 mM DTT]. Columns were rotated overnight at 4 °C. Beads were collected by gentle centrifugation and washed three times for 10 min with ice-cold binding buffer before eluting bound proteins with 30 µL of Laemmli loading buffer pre-heated at 90 °C. 0.5% of the samples (eluted material or flow-through) were run onto 8% SDS-polyacrylamide gels that were transferred to nitrocellulose membranes which were reacted with α-His tag antibody (#631212, Clontech). Quantitation of band intensities was done with ImageLab software (BioRad). In parallel, 4% of the elution sample volume was loaded onto 10% SDS-polyacrylamide gel and stained with coomassie dye (BlueSafe, NZY) to confirm recovery of SNARE-GST baits.

## Pull-down of TNT-expressed Sly1-HA3

Sly1-HA3 was synthesized with the TNT SP6 Quick Coupled Transcription/Translation system (#L2080, Promega), according to the instructions of the manufacturer. The reaction was primed with 1 µg of pSP64::Sly1-HA3 cDNA. 10 µL of the resulting mix were combined with 15 µL of glutathione-Sepharose beads, previously loaded with SNARE-GST baits as described above, in 0.4 ml of pull-down binding buffer, using 0.8 ml Pierce columns that were rotated overnight at 4 °C before beads and flow-through were recovered after gentle centrifugation. Beads were washed three times for 10 min at 4 °C in pull-down binding buffer before eluting bound material with 30 µL of Laemmli loading buffer for 2 min at 90 °C. 20% of the elution sample volume was analyzed by western blotting with α-HA tag antibody (#3F10, Roche) for Sly1-HA immunodetection.

## Size exclusion chromatography of HA-tagged cell extracts

Gel filtration experiments were performed as described (*Bravo-Plaza et al., 2019*). Briefly, 200 µL of cell extract were loaded onto a Superose 6 10/300 column (Pharmacia) equilibrated with running buffer [25 mM Tris-HCl (pH 7.5), 600 mM KCl, 4 mM EDTA, 1 mM DTT]. Fractions of 0.5 ml were collected, from which 80 µL were mixed with 40 µL of Laemmli loading buffer and denatured at 90 °C. 25 µL of these samples were resolved by SDS-PAGE and analyzed by western blotting with α-HA3 tag antibody (#3F10, Roche) for western blotting. Sizing standards were myoglobin (17 kDa), BSA (67 kDa), aldolase (158 kDa) ferritin (449 kDa), thyroglobulin (669 kDa), and dextran blue (Vo).

## Analytical ultracentrifugation: sedimentation velocity

Sedimentation velocity assays and subsequent raw data analysis were performed in the Molecular Interactions Facility of the Centro de Investigaciones Biológicas Margarita Salas. Samples (320 µL) in PBS containing 5% (v/v) glycerol and 1 mM β-mercaptoethanol were loaded into analytical ultracentrifugation cells, which were run at 20 °C and 48,000 rpm in a XL-I analytical ultracentrifuge (Beckman-Coulter Inc) equipped with UV-VIS absorbance and Raleigh interference detection systems, using an An-50Ti rotor, and 12 mm Epon-charcoal standard double-sector centerpieces. Sedimentation profiles were recorded at 230 nm. Differential sedimentation coefficient distributions were calculated by least-squares boundary modeling of sedimentation velocity data using the continuous distribution $c(s)$ Lamm equation model as implemented by SEDFIT (*Schuck, 2000*). Experimental Svedberg coefficient values were corrected to standard conditions ($s_{20,w}$: water, 20 °C, and infinite dilution) using SEDNTERP software.

## Dynamic light scattering (DLS)

DLS experiments were carried out in a Protein Solutions DynaPro MS/X instrument at 20 C using a 90° light scattering cuvette. DLS autocorrelation functions, an average of at least 18 replicates, were collected with Dynamics V6 software. Analysis evidenced in most cases the presence of two diffusing species, one with faster diffusion corresponding to a discrete major species and a second with substantially slower diffusion, corresponding to higher order species, with a minor contribution to the whole population within the sample. Exceptions were the globular domain GHD that appeared as a single species, and the coiled-coil construct with a larger contribution of the higher-order species. Dynamics software was also employed to export the data as text files for parallel analysis using user-written scripts and functions in MATLAB (Version 7.10, MathWorks, Natick, MA). A double exponential decay model was fit to the data *via* nonlinear least squares, using as starting values the translational diffusion coefficients and relative amounts of the two different species, using as starting values those from the regularization analysis and their masses (those of the discrete species as estimated by the Svedberg equation). These values were compatible with the experimental data, rendering a similar best-fit value of the diffusion coefficient for the major species and allowing to assess its probability distribution.

## Estimate of molar mass from hydrodynamic measurements

The apparent molar masses of Uso1 and its mutants were calculated via the Svedberg equation, using the *s*- and *D*-values of the major species independently measured by sedimentation velocity and DLS, respectively.

## Fluorescence microscopy

*A. nidulans* hyphae were cultured in Watch Minimal Medium WMM (*Peñalva, 2005*). Image acquisition equipment, microscopy culture chambers, and software have been detailed (*Pinar et al., 2022*; *Pinar and Peñalva, 2020*). Simultaneous visualization of green and red emission channels was achieved with a Gemini Hamamatsu beam splitter coupled to a Leica DMi8 inverted microscope. Z-Stacks were deconvolved using Huygens Professional software (version 20.04.0p5 64 bits, SVI). Images were contrasted with Metamorph (Molecular Devices). Statistical analysis was performed with GraphPad Prism 8.02 (GraphPad). Uso1-GFP time of residence in cisternae was estimated from 3D movies consisting of middle planes with 400 photograms at 2 fps time resolution. Each Uso1 puncta considered in the analysis was tracked manually with 3D (x, y, t) representations generated with Imaris

software (Oxford Instruments) combined with direct observation of photograms in movies and kymograph representations traced across >25 px-wide linear ROI covering the full width of the hyphae.

| Antibody | Dilution | Origin | Reference |
|---|---|---|---|
| **Primary** | | | |
| α-HA | 1:1000 | rat | 3F10 clone, Roche |
| α-His6x tag | 1:10,000 | mouse | #631212, Clontech |
| α-tubulin | 1:5000 | mouse | DM1A clone, Sigma |
| α-Uso1 | 1:1000 | rabbit | Polyclonal antiserum, Davids Biotechnologie |
| **Secondary (HRP-conjugated)** | | | |
| α-rat IgG | 1:4000 | goat | #3010–05, Southern Biotech |
| α-mouse IgG | 1:5000 | goat | #A9044, Sigma |
| α-rabbit IgG | 1:2000 | donkey | #NA934, Amersham |

## AlphaFold predictions

AlphaFold2 (*Jumper et al., 2021*) predictions were run using versions of the program installed locally and on ColabFold (*Mirdita et al., 2022*) with the AlphaFold2_advanced.ipynb notebook and the MMseqs2 MSA option. In all cases, the five solutions predicted by AlphaFold2 by default were internally congruent, and we always chose the one ranked first by the software. Uso1 GHD (1-674), Bos1, Bet1, and RAB1 were initially submitted as hetero-oligomers. Subsequently, predictions were submitted as 1:1 complexes of Uso1 GHD with Bos1, Bet1, and RAB1, as described in the table below. The solutions were also very similar when comparing the different combinations displayed in the table that were fed to the software, strongly supporting the validity of the results.

| AlphaFold2 runs, subunits included | | | |
|---|---|---|---|
| Uso1 GHD | Bet1 | | |
| Uso1 GHD | Bet1 | Bos1 | |
| Uso1 GHD | Bet1 | Bos1 | Rab1 |
| Uso1 GHD | Bet1 (1-49) | | |
| Uso1 GHD (N-terminal) | Bet1 (1-49) | | |
| Uso1 GHD | Bet1 (1-49) | Bos1 Habc (1-126) | |
| Uso1 GHD E6K/G540S | Bet1 (1-49) | Bos1 Habc (1-126) | |
| Uso1 GHD | | Bos1 | |
| Uso1 GHD | | Bos1 Habc (1-126) | |
| Uso1 GHD dimer | | Bos1 Habc (1-126) | |
| Uso1 GHD E6K/G540S | | Bos1 | |
| Uso1 GHD E6K/G540S | | Bos1 Habc (1-126) | |
| Uso1 | | | |
| Uso1 dimer | | | |
| Uso1 dimer E6K/G540S | | | |

*Continued on next page*

*Continued*

| AlphaFold2 runs, subunits included | | | |
|---|---|---|---|
| Bet1 | Bos1 | Sec22 | Sed5 |
| Rud3 dimer | | | |
| Coy1 dimer | | | |
| Grh1 | Bug1 | | |

## Availability statement

All DNA molecules used here may reconstructed by standard techniques using primers listed in *Supplementary file 3*. All strains listed under *Supplementary file 2* are available for academic purposes upon reasonable request to the corresponding author. They are deposited and maintained by the corresponding laboratory in the so-denoted Madrid (MAD) collection.

## Acknowledgements

We thank Juan R Luque (Molecular Interactions Facility, Centro de Investigaciones Biológicas) for his help with the analytical ultracentrifuge experiments, Manuel Sánchez-Berges for HA3-tagged SNARE strains, Sara Abib and Elena Reoyo for skillful technical assistance. Thanks are due to Spain's Ministerio de Ciencia e Innovación for grants RTI2018-093344-B100 (MAP) and predoctoral contract BES-2016–077440 (IB.-P/MAP), and to the Comunidad de Madrid for grant S2017/BMD-3691 (MAP). Grants were co-funded by European Regional Development and European Social Funds. The authors declare that they do not have any competing financial interests.

## Additional information

### Funding

| Funder | Grant reference number | Author |
|---|---|---|
| Agencia Estatal de Investigación | BES-2016-077440 | Ignacio Bravo-Plaza |
| Agencia Estatal de Investigación | RTI2018-093344-B100 | Miguel A Peñalva |
| Comunidad Autónoma de Madrid | S2017/BMD-3691 | |
| European Regional Development Fund | | |
| European Social Fund | | |
| Agencia Estatal de Investigación | PID2021-124278OB-I00 | |

The funders had no role in study design, data collection and interpretation, or the decision to submit the work for publication.

### Author contributions

Ignacio Bravo-Plaza, Conceptualization, Data curation, Formal analysis, Supervision, Validation, Investigation, Visualization, Writing – review and editing; Victor G Tagua, Data curation, Formal analysis, Validation, Investigation; Herbert N Arst, Conceptualization, Data curation, Formal analysis, Supervision, Validation, Investigation, Writing – review and editing; Ana Alonso, Data curation, Formal analysis, Investigation, Methodology, Project administration; Mario Pinar, Conceptualization, Data curation, Formal analysis, Supervision, Validation, Investigation, Methodology, Writing – review and editing; Begoña Monterroso, Conceptualization, Data curation, Formal analysis, Supervision, Validation, Writing – review and editing; Antonio Galindo, Conceptualization, Data curation, Formal analysis,

Validation, Investigation, Visualization, Methodology, Writing – review and editing; Miguel A Peñalva, Conceptualization, Data curation, Formal analysis, Supervision, Funding acquisition, Validation, Investigation, Visualization, Writing – original draft, Writing – review and editing

### Author ORCIDs
Ignacio Bravo-Plaza  http://orcid.org/0000-0003-0934-9084
Victor G Tagua  http://orcid.org/0000-0003-1494-6895
Mario Pinar  http://orcid.org/0000-0002-2415-8721
Begoña Monterroso  http://orcid.org/0000-0003-2538-084X
Miguel A Peñalva  http://orcid.org/0000-0002-3102-2806

### Decision letter and Author response
Decision letter https://doi.org/10.7554/eLife.85079.sa1
Author response https://doi.org/10.7554/eLife.85079.sa2

## Additional files

### Supplementary files
- Supplementary file 1. 'Protein interfaces, surfaces, and assemblies' service PISA.
- Supplementary file 2. List and complete genotypes of *A. nidulans* strains used in this work.
- Supplementary file 3. Primers used for PCR-based genetic manipulations.
- MDAR checklist

### Data availability
All data generated or analysed during this study are included in the manuscript and supporting file; Source Data files have been provided for Figures 8 through 11, 13 -figure supplement 2 and 14Supplementary Table II and III(strains and primers) ensure the reproducibility of the experiments.

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
