## [Editor Report]

This valuable manuscript explores the role of Uso1/p115, a protein that has been implicated in vesicle tethering at the ER-Golgi interface. By investigating a Uso1 mutant that allows Aspergillus cells to survive in the absence of Rab1, the authors conclude that the essential role of Uso1 is not actually tethering, but rather SNARE complex assembly mediated by the globular head domain. This convincing analysis significantly advances our understanding of Uso1 and also prompts a reevaluation of long-standing assumptions about coiled-coil proteins involved in vesicular transport.

---

## [Decision Letter]

**Decision letter after peer review:**

Thank you for submitting your article "Tethering by Uso1 is dispensable: The Uso1 monomeric globular head domain interacts with SNAREs to maintain viability." for consideration by *eLife*. Your article has been reviewed by 3 peer reviewers, and the evaluation has been overseen by a Reviewing Editor and Suzanne Pfeffer as the Senior Editor. The following individuals involved in the review of your submission have agreed to reveal their identity: Benjamin S Glick (Reviewer #1); Giulia Zanetti (Reviewer #2).

All three reviewers agree that this study is well executed and that it represents a significant advance. The text does not express it this way, but a reasonable interpretation is that for the ER-Golgi SNARE complex, Uso1 plays a role analogous to those of the multi-subunit tethers that promote the assembly of other SNARE complexes. If the authors agree, perhaps this point could be emphasized.

The reviewers have a number of concerns and recommendations. Among the suggestions are simple experiments, improvements to the presentation, and more cautious interpretations of some of the modeling and experimental data.

Rather than stipulating which revisions are essential, please go through the reviews carefully and either address each point or explain in a cover letter why that aspect of the manuscript should remain unchanged. Issues the reviewers considered important included:

1) The Coy1/Rud3 section adds little and is distracting.

2) The novelty of the conclusions should be explained more clearly in light of previous evidence that Rab1 and Uso1 can be bypassed by enhancing SNARE assembly.

3) The evidence that the globular head domain is a monomer in vivo is not definitive, so this conclusion should be tempered.

4) Modeling the coiled coil is not beneficial and may be misleading.

5) Evidence should be provided that Golgi structures are normal in rab1-ts cells until restored by Uso1 E6K/G540S.

We look forward to seeing a revised manuscript.

*Reviewer #1 (Recommendations for the authors):*

1. In line 108: I'm not sure it's accurate to state that "most Golgi tethers are functionally redundant." Most of them are nonessential, but each one probably has a unique role.

2. Does "CCD" stand for a coiled-coil domain? If so, it's incorrect to write "CCD domain" (e.g., line 216).

3. The analysis of Coy1 and Rud3 is a digression that does not seem to add to the story. This section disrupts the flow of the narrative and could be removed.

*Reviewer #2 (Recommendations for the authors):*

Required revisions:

Line 178: remove: 'uncoated". This is not what the cited paper claims and other work (e.g. Ferro-Novik lab) showed uncoating happens after uso1 is recruited, at least in fungi.

Figure 2: showing a domain diagram would be helpful (like that in Figure 3), but with the correct colours.

Figure 2: I don't see the point in modelling the coiled-coil. It suggests a particular overall architecture and positioning of the various segments that have similar chances of being right as well as wrong. Showing the model of the coiled-coil region does not add anything to the story and might just be misleading. I'd be more comfortable with models made with the N-terminal and GDH domains only. The fact that the C-term is forming coiled coils can be shown diagrammatically without proposing an atomic model.

Section: "coiled-coil mediated dimerization of Uso1: the globular head is monomeric":

The authors suggest a conformational switch in Uso1 that is mediated by the positive head interacting with the negative tail, but +ve charges on the head domain are not shown/mentioned.

Also, the potential role of his tag (which is not cleaved in these experiments) is not discussed. The authors discount the possibility that Uso1 adopts a closed conformation that is disrupted by the mutations, but the His tag could very well be inhibiting 'closure' as it would imply proximity of positive with positive regions. It cannot be excluded that this is the reason why they don't see changes unless the experiment is done after his-tag cleavage.

In general, I think experiments around Figure 3 are useful for checking monomer versus dimer conformation but the interpretation in terms of conformational change is less clear. As I don't think it is a crucial argument to their story it would probably strengthen the paper to remove reference to conformational changes.

Figure 3 would benefit from a schematic diagram in each panel rather than the current top panel.

Line 339-341: " in view of this…". I actually don't see why the data discussed indicates that Uso1 puncta originate downstream of COPII. Either clarify or remove.

Line 346: likewise, the data discussed above does not indicate Uso1 is on a membrane with Golgi identity. This is established later in the paper, so it isn't wrong, but the logical flow is poor.

Section: "the globular head domain of Uso1 carrying the double E6K/G540S substitution supports cell viability"

Regarding the absence of dimerization of GDH domain: the data is convincing and valid, but does not allow us to definitely conclude that GDH is a monomer in situ. It could be dimerising transiently/weekly, not allowing detection in vitro. In fact, alphafold seems to suggest a dimerization interface with high confidence.

Throughout the manuscript, the statements that dimerization is not happening/not needed in vivo should be toned down as it might still occur at the site of GDH binding to vesicles.

Figure 9: It is surprising that overexpressed WT Uso1 supports viability more than the mutant. The authors suggest it is due to the toxicity of mutants due to high expression and tight binding. But if this was the case, why would the same effect not be seen in the Uso+ background?

Section: "Golgi SNAREs bind directly to the Uso1 GHD; effects of Uso1 E6K/G540S"

The authors nicely show that G540 is at the interface between Uso1 and Bos1, providing a nice complementation between alphafold2 models and their experimental results. They also show how the G540S mutant affects the interface. It would nicely complete the analysis if the models obtained with WT and G504S mutants could be run through the PISA PDB server to calculate binding energies and confirm the tighter binding predicted for the mutant.

Figure 12: the alphafold2 structures coloured by confidence seem to be missing for the Uso1-Bos1 complexes. They should be added as they are the basis for a main conclusion.

The colour code in figures is poorly chosen: the same proteins, especially the main focus of the paper should be depicted with a consistent colour throughout. Even when not possible, at least self-consistency within the same figure should be obtained.

*Reviewer #3 (Recommendations for the authors):*

Suggestions to strengthen:

1. Figure 7 shows that expression of GFP-tagged Uso1 E6K/G540S restores localization of Uso1 to Golgi punctate structures in rab1ts cells. But it was not clear if the Golgi punctate structures are normal in rab1ts cells. Or are Golgi structures dispersed in rab1ts cells until restored by Uso1 E6K/G540S? Imaging a Golgi marker (e.g. Sed5) in rab1ts cells would strengthen the proposal that the E6K/G540S mutations increase Uso1 localization to normal Golgi structures in rab1ts cells.

2. On Page 19, regarding AlphaFold predictions, it is stated that "Lys (as in E6K) reinforces the positive charge of this a-helix, which would be inserted into the vesicle membrane, potentially contributing to Uso1 recruitment to COPII vesicles." This statement would be strengthened if it could be shown that the single E6K mutant displays increased levels of membrane association compared to the wild-type protein. Presumably, this association would be SNARE protein independent.

3. Based on the AlphaFold model in Figure 12, can it be demonstrated that the single G540S mutant is sufficient for increased binding to Bos1-GST? Similarly, do mutations in specific residues of Bos1 and Bet1 that are predicted to interact with surface residues of Uso1 GHD decrease binding? This could strengthen the proposed molecular model.

4. It would be helpful to explain why the Bet1-GST protein used in these studies deletes the SNARE motif while the other SNARE-GST proteins include their full cytoplasmic domains and SNARE motif.

---

## [Author Response]

The reviewers have a number of concerns and recommendations. Among the suggestions are simple experiments, improvements to the presentation, and more cautious interpretations of some of the modeling and experimental data.Rather than stipulating which revisions are essential, please go through the reviews carefully and either address each point or explain in a cover letter why that aspect of the manuscript should remain unchanged. Issues the reviewers considered important included:

As far as your recommendations are concerned we have taken the following actions:

0.Uso1 resembling MTCs?

We have emphasized, expanding the "ideas and speculation" section, that Uso1 might behave as a multisubunit tethering complex, templating SNARE assembly.

1) The Coy1/Rud3 section adds little and is distracting.

We have removed the Coy1/Rud3 section as suggested.

2) The novelty of the conclusions should be explained more clearly in light of previous evidence that Rab1 and Uso1 can be bypassed by enhancing SNARE assembly.

This comment led us to rewrite the discussion, the ideas were there but they were not well structured.

3) The evidence that the globular head domain is a monomer in vivo is not definitive, so this conclusion should be tempered.

We have tempered this conclusion in the discussion, but we noted that, as the referee picked up, that a key table was omitted in the incomplete version of figure 3 that we unfortunately submitted. Details are given in the answer to referee number two.

4) Modeling the coiled coil is not beneficial and may be misleading.

We have modified former figure 2A, but we should like to keep the modelling of the coiled coil, because we want readers to appreciate visually the relative sizes of the GHD and the coiled coil. To avoid misleading readers, we have added a cautionary statement in the text.

5) Evidence should be provided that Golgi structures are normal in rab1-ts cells until restored by Uso1 E6K/G540S.

This was one of the major improvements introduced in this revised version.

Reviewer #1 (Recommendations for the authors):1. In line 108: I'm not sure it's accurate to state that "most Golgi tethers are functionally redundant." Most of them are nonessential, but each one probably has a unique role.2. Does "CCD" stand for a coiled-coil domain? If so, it's incorrect to write "CCD domain" (e.g., line 216).3. The analysis of Coy1 and Rud3 is a digression that does not seem to add to the story. This section disrupts the flow of the narrative and could be removed.

In his only general comment, this referee asked for our acknowledging that the correlation that we find between stronger SNARE binding in our mutants and rescue of rab1∆ is not a formal demonstration that suppression occurs by augmented binding. He is quite right. We have introduced a complete paragraph (P 23, starting line 817) to address this point, thank you.

Reviewer #2 (Recommendations for the authors):Required revisions:Line 178: remove: 'uncoated". This is not what the cited paper claims and other work (e.g. Ferro-Novik lab) showed uncoating happens after uso1 is recruited, at least in fungi.

Fone, thank you.

Figure 2: showing a domain diagram would be helpful (like that in Figure 3), but with the correct colours.Figure 2: I don't see the point in modelling the coiled-coil. It suggests a particular overall architecture and positioning of the various segments that have similar chances of being right as well as wrong. Showing the model of the coiled-coil region does not add anything to the story and might just be misleading. I'd be more comfortable with models made with the N-terminal and GDH domains only. The fact that the C-term is forming coiled coils can be shown diagrammatically without proposing an atomic model.Section: "coiled-coil mediated dimerization of Uso1: the globular head is monomeric":The authors suggest a conformational switch in Uso1 that is mediated by the positive head interacting with the negative tail, but +ve charges on the head domain are not shown/mentioned.

We understand the concerns of the referee, but we would like to keep it there because the modelling of the CCD didn't attempt to provide a definitive structure. Its role was that readers could appreciate visually the actual length of the coiled coil relative to the globular domain. However, in view of the comment, we have introduced a cautionary statement, indicating readers that the prediction of the coiled-coil is presented with the only purpose of comparing visually the sizes of the coiled coil domain and the globular domain.

Also, the potential role of his tag (which is not cleaved in these experiments) is not discussed. The authors discount the possibility that Uso1 adopts a closed conformation that is disrupted by the mutations, but the His tag could very well be inhibiting 'closure' as it would imply proximity of positive with positive regions. It cannot be excluded that this is the reason why they don't see changes unless the experiment is done after his-tag cleavage.

Yes, we do, because the experiments of velocity sedimentation are not compatible with that. This comment is possibly the result of a mistake in our submission, as the key summary table that was attached to this figure was omitted, and the referee could not appreciate the quality of the data.

In general, I think experiments around Figure 3 are useful for checking monomer versus dimer conformation but the interpretation in terms of conformational change is less clear. As I don't think it is a crucial argument to their story it would probably strengthen the paper to remove reference to conformational changes.

We respectfully disagree with the referee. As the reviewer points out, sedimentation velocity experiments are very useful for the determination of the association states of species. But sedimentation velocity data also disclose conformational changes, as they contain information about the shape of particles (Lebowitz 2002). The *s*-value is determined, among other molecular parameters as defined by the Svedberg equation (Svedberg, 1940), by the sedimentation frictional coefficient of the species, which informs about the shape of a molecule. For a given mass, a globular molecule – close to a compact sphere – is the fastest possible sedimenting species (i.e. sediments with an *s* close to the maximum), because it has the minimum surface area in contact with the solvent and, hence, the frictional coefficient is the minimum (Lebowitz 2002). In other words, for a given mass, a particle with an extended shape, because of the larger frictional coefficient, sediments slower than a globular particle. Examples of conformational changes characterized by AUC can be found in (Schuck 2001, Matte 2012, Brautigam 2020). Our statement that the double mutation in full-length Uso1 does not induce large conformational changes (lines 259 …) perfectly holds, as they have virtually the same mass and the same *s*-value. In the case of the comparisons of GHD (3.7S, 70 kDa) with CCD (2.7S, 100 kDa; lines 268-269) and with full length Uso1 (4.8S, 250 kDa; lines 271-276), the *s*-values are, in all likelihood, related with an asymmetry of the last two, which for the complete Uso1 has been demonstrated documented by EM. We have slightly rephrased the statement in line 284 to tone down the message.

Brautigam et al. 2020. Using modern approaches to sedimentation velocity to detect conformational changes in proteins. European Biophysics Journal 49:729–743.Lebowitz J. et al. 2002. Modern analytical ultracentrifugation in protein science: A tutorial review. Protein Science, 11:2067–2079.Matte et al. 2012. Characterization of Conformational Changes and Protein-Protein Interactions of Rod Photoreceptor Phosphodiesterase (PDE6). JBC 287:20111–20121.Schuck P,et al., 2001. Rotavirus Nonstructural Protein NSP2 Self-assembles into Octamers That Undergo Ligand-induced Conformational Changes. JBC, 276:9679–9687.Svedberg T., Pedersen K.O. 1940. The Ultracentrifuge, Oxford University Press, London.

There is an additional point to consider: in figure 9B, we show that in a gel filtration is experiment the GHD isolated from Aspergillus cells elutes at the same position than the protein purified from bacteria, and that this position corresponds to a monomer (co-eluting with 67 kDa globular bovine albumin).

His tag interfering with close conformation: Uso1 adopts a closed conformation that is disrupted by the mutations, but the His-tag keeps them permanently open by electrostatic interactions between N- Terminal acidic residues and C-terminal histidines: it seems unlikely that E6 and K6 Uso1 behave similarly in this particular regard. Please, also note that on figure 9B we show that full-length Uso1 shows the same elution profile in a gel filtration column irrespective of whether the protein is C-terminally tagged with His6 or HA3,

Figure 3 would benefit from a schematic diagram in each panel rather than the current top panel.

Schematic diagrams have been introduced as suggested by the referee.

Puncta originate downstream of COPII, removed as suggested.

Line 339-341: “ in view of this…”. I actually don’t see why the data discussed indicates that Uso1 puncta originate downstream of COPII. Either clarify or remove.Line 346: likewise, the data discussed above does not indicate Uso1 is on a membrane with Golgi identity. This is established later in the paper, so it isn’t wrong, but the logical flow is poor.

Corrected on line 353.

Section: “the globular head domain of Uso1 carrying the double E6K/G540S substitution supports cell viability”.Regarding the absence of dimerization of GDH domain: the data is convincing and valid, but does not allow us to definitely conclude that GDH is a monomer in situ. It could be dimerising transiently/weekly, not allowing detection in vitro. In fact, alphafold seems to suggest a dimerization interface with high confidence.Throughout the manuscript, the statements that dimerization is not happening/not needed in vivo should be toned down as it might still occur at the site of GDH binding to vesicles.Figure 9: It is surprising that overexpressed WT Uso1 supports viability more than the mutant. The authors suggest it is due to the toxicity of mutants due to high expression and tight binding. But if this was the case, why would the same effect not be seen in the Uso+ background?Section: "Golgi SNAREs bind directly to the Uso1 GHD; effects of Uso1 E6K/G540S"The authors nicely show that G540 is at the interface between Uso1 and Bos1, providing a nice complementation between alphafold2 models and their experimental results. They also show how the G540S mutant affects the interface. It would nicely complete the analysis if the models obtained with WT and G504S mutants could be run through the PISA PDB server to calculate binding energies and confirm the tighter binding predicted for the mutant.

The globular head is monomeric? Our data indicate that the GHD is a monomer whether extracted from bacteria or from Aspergillus. The referee argues that dimerization could occur in situ. It is difficult to argue against that, but we note that the coiled-coil has been completely removed from our construct. Moreover, figure 3-supplemental figure 1 shows that the GHD is monomeric at concentrations up to 5 µM, which appears incompatible with the presence of an efficient dimerization surface. Nevertheless, to avoid the concerns of the referee, we have toned down this point across the whole manuscript.

Overexpression and differences in toxicity: to be honest we do not have a straightforward explanation and we prefer not to speculate to avoid statements that might draw the readers' attention away from important points of the manuscript. Given the complexity of these macromolecular machines, the referee will surely appreciate that understanding this paradox might be a project by itself.

PISA analysis: as the reviewer suggests, a PISA analysis was run and mentioned in the former version of the manuscript. Under "mechanistic insights… " we provided some of the data pointing out the increment in surface of interaction resulting from G540S. We have now expanded information by adding a summary of the results as a new supplemental table 1. As already noted before, there is a discrete increment in the surface area from 824.5Å2 in the model for the wild-type complex to 833.1Å2 for the mutant. The change of G540 to S creates four new hydrogen bonds. Roughly, each hydrogen bond contributes about 0.5 kcal/mol into the free energy, which, combined with the ~0.3 kcal/mol of the four salt bridges on each of the models, makes an approximately total binding energy of -8.4 kcal/mol for the wild type complex and -9.6 kcal/mol for the mutant one (second complete paragraph under interpretation of the mutant, p 19-20)

Figure 12: the alphafold2 structures coloured by confidence seem to be missing for the Uso1-Bos1 complexes. They should be added as they are the basis for a main conclusion.

Uso1::Bos1 complex, predicted error now on figure 12, figure 3 supplement.

The colour code in figures is poorly chosen: the same proteins, especially the main focus of the paper should be depicted with a consistent colour throughout. Even when not possible, at least self-consistency within the same figure should be obtained.

Color codes now consistent & minor changes corrected.

Reviewer #3 (Recommendations for the authors):Suggestions to strengthen:1. Figure 7 shows that expression of GFP-tagged Uso1 E6K/G540S restores localization of Uso1 to Golgi punctate structures in rab1ts cells. But it was not clear if the Golgi punctate structures are normal in rab1ts cells. Or are Golgi structures dispersed in rab1ts cells until restored by Uso1 E6K/G540S? Imaging a Golgi marker (e.g. Sed5) in rab1ts cells would strengthen the proposal that the E6K/G540S mutations increase Uso1 localization to normal Golgi structures in rab1ts cells.2. On Page 19, regarding AlphaFold predictions, it is stated that "Lys (as in E6K) reinforces the positive charge of this a-helix, which would be inserted into the vesicle membrane, potentially contributing to Uso1 recruitment to COPII vesicles." This statement would be strengthened if it could be shown that the single E6K mutant displays increased levels of membrane association compared to the wild-type protein. Presumably, this association would be SNARE protein independent.3. Based on the AlphaFold model in Figure 12, can it be demonstrated that the single G540S mutant is sufficient for increased binding to Bos1-GST? Similarly, do mutations in specific residues of Bos1 and Bet1 that are predicted to interact with surface residues of Uso1 GHD decrease binding? This could strengthen the proposed molecular model.4. It would be helpful to explain why the Bet1-GST protein used in these studies deletes the SNARE motif while the other SNARE-GST proteins include their full cytoplasmic domains and SNARE motif.

Novelty of our conclusions? We have completely re-written the discussion to emphasize the contribution of our work; note that we have added more, we believe sound, experiments. Due to that reason, the abstract has also been fully re-written as has been the discussion.

Membrane targeting by E6K. We are very grateful to the referee for noting the lack of evidence supporting the interpretation that the N-terminal helix would bind membranes. It appears that it does not. We have removed from the manuscript any mention to this possibility (see below).

Are Golgi structures affected my rab1ts, and do they recover by the presence of E6K/G540S Uso1? This was a very nice suggestion. We chose GeaA/Gea1-mCh to label early Golgi cisternae and crossed it into the Uso1-GFP backgrounds. The experiment worked beautifully and has now been included as Figure 7C and 7D. rab1ts results in an statistically significant decrease in the number of cisternae and an increase in the cytosolic haze which is remediable by the additional presence of double mutant Uso1.

A collateral effect of this comment is that we were not happy with the incomplete colocalization of Uso1 with Golgi markers. We hypothesized that Uso1 would localise in a very early Golgi compartment. Then we used Rer1, a cargo receptor that cycles continuously between the Golgi and the ER to label this "very early" Golgi and we found essentially complete co-localization. This conclusion adds to the conclusions of the paper.

Analysis of single mutants. We carried out this experiment. G540S was indeed capable of providing by itself strong binding to Bos1-GST. E6K was also capable of increasing the binding to the SNARE significantly, although, in agreement with the respective phenotypes, to a lesser extent than G540S. Therefore, it seems that E6K contributes to rescue rab1∆ by binding to SNAREs. These experiments have been included as panels G and H in figure 11.

Bet1-GST; There was a mistake in the coordinates of this construct in our list of plasmids. Indeed, we tried to split the Bet1 cytosolic region into SNARE and non-SNARE domains and the polypeptides were quite unstable. In all experiments included in the paper, the complete cytosolic domain of the SNAREs was used. We have added a statement to the Materials and methods.